# Unified Breakdown Analysis for Byzantine Robust Gossip

**Renaud Gaucher** [1 2]  **Aymeric Dieuleveut** [1]  **Hadrien Hendrikx** [2]

## Abstract

In decentralized machine learning, different devices communicate in a peer-to-peer manner to collaboratively learn from each other's data. Such approaches are vulnerable to misbehaving (or Byzantine) devices. We introduce F–RG, a general framework for building robust decentralized algorithms with guarantees arising from robust-sum-like aggregation rules F. We then investigate the notion of *breakdown point*, and show an upper bound on the number of adversaries that decentralized algorithms can tolerate. We introduce a practical robust aggregation rule, coined $CS_+$, such that $CS_+$–RG has a near-optimal breakdown. Other choices of aggregation rules lead to existing algorithms such as $\mathrm{ClippedGossip}$ or NNA. We give experimental evidence to validate the effectiveness of $CS_+$–RG and highlight the gap with NNA, in particular against a novel attack tailored to decentralized communications.

## 1. Introduction

Distributed machine learning, in which the training process is performed on multiple computing units (or nodes), responds to the increasingly distributed nature of data, its sensitivity, and the rising computational cost of optimizing models. We investigate the decentralized paradigm of distributed learning, in which nodes communicate in a peer-to-peer manner within a communication network, in opposition to distributed architectures relying on a central server that coordinates all units.

Distributing the training over a large number of devices introduces new security issues: software may be faulty, local data may be corrupted, and nodes can be hacked or even controlled by a hostile party. Such issues are modeled as *Byzantine* node failures (Lamport et al., 1982), defined as omniscient adversaries able to collude with each other. Standard distributed learning methods are known to be vulnerable to these Byzantine attacks (Blanchard et al., 2017). This has led to significant efforts in the development of robust distributed learning algorithms. From the first works on Byzantine-robust SGD (Blanchard et al., 2017; Yin et al., 2018; Alistarh et al., 2018; El-Mhamdi et al., 2020), methods have been developed to tackle stochastic noise using Polyak momentum (Karimireddy et al., 2021; Farhadkhani et al., 2022) and mixing strategies to handle heterogeneous loss functions (Karimireddy et al., 2023; Allouah et al., 2023). In parallel to these robust algorithms, efficient attacks have been developed to challenge Byzantine-robust algorithms (Baruch et al., 2019; Xie et al., 2020). Recently, (Allouah et al., 2025) have used pre-aggregation adaptive clipping to improve the robustness. To bridge the gap between algorithm performance and achievable accuracy in the Byzantine setting, tight lower bounds have been constructed for the heterogeneous setting (Karimireddy et al., 2023; Allouah et al., 2024). Yet, all these works rely on a trusted central server to coordinate the training.

In contrast, the decentralized case has been less explored. Especially when the communication network, abstracted as a graph where vertices represent computing units that communicate through edges, is not fully connected (a.k.a. sparse). For instance, understanding *how many Byzantine nodes can be tolerated over an arbitrary network before a communication protocol fails* is an open question (that we address in this paper). Indeed, the network is often assumed to be fully connected (El-Mhamdi et al., 2021; Farhadkhani et al., 2023). If the work of (Fang et al., 2022) addresses the case of sparse networks, they only consider homogeneous losses. Closer to our setting, He et al. (2023); Wu et al. (2023) tackle Byzantine optimization on sparse networks with heterogeneous losses, however, they only achieve suboptimal robustness, as their guarantees vanish with either the number of nodes or the connectivity of the network. Moreover, the communication scheme proposed in He et al. (2023) relies on inaccessible information. Similarly, there is a lack of attacks designed to challenge decentralized optimization. Up to our knowledge, only He et al. (2023) propose one, named *Dissensus*. Still, a few previous works

[1]Centre de mathématiques appliquées, École polytechnique, Institut Polytechnique de Paris, Palaiseau France [2]Centre Inria de l'Univ. Grenoble Alpes, CNRS, LJK, Grenoble, France. Correspondence to: Renaud Gaucher <renaud.gaucher@polytechnique.edu>.

*Proceedings of the 42^{nd} International Conference on Machine Learning*, Vancouver, Canada. PMLR 267, 2025. Copyright 2025 by the author(s).

(Wu et al., 2023; Farhadkhani et al., 2023) have investigated generic criteria to go from communication schemes to robust distributed SGD frameworks.

Our work proposes a generic method to design algorithms that solve the aforementioned shortcomings. To do so, we carefully study the decentralized mean estimation problem. This seemingly simple problem retains most of the difficulty of handling Byzantine nodes while allowing us to derive strong convergence and robustness guarantees. Our solution relies on robust adaptations of the *gossip* communication, a popular scheme for decentralized communication. We then tackle general (smooth non-convex) optimization problems through a reduction. Our contributions are the following:

**1 - Unifying algorithmic framework.** We develop a generic method (the RG method) to construct and analyze robust communication algorithms. It is based on the decentralized application of robust aggregation rules. This RG method recovers NNA (Farhadkhani et al., 2022) and ClippedGossip (He et al., 2023) in specific cases. We use the RG method to build $CS_+$–RG, a novel communication algorithm that is both practical and adapted to a sparse communication network.

**2 - Tight theoretical guarantees.** We show that RG provides robust convergence guarantees as soon as the underlying aggregation rule verifies $(b, \rho)$–robustness, a new robustness criterion that we introduce, and the weight of Byzantine nodes is not too large (with respect to the *algebraic connectivity*, a spectral property of the communication graph). We also show the converse result, that is, no robustness guarantees can be obtained if the weight of Byzantine nodes exceeds this threshold. Our bounds match each other for specific aggregation rules. These results generalize the standard fully-connected breakdown point of $1/3$ of Byzantines nodes to arbitrary sparse networks.

Besides these core contributions, we introduce a theoretically grounded attack, called *Spectral Heterogeneity*, specifically designed to challenge decentralized algorithms by leveraging spectral properties of the communication graph.

The remainder of the paper is organized as follows. Section 2 formalizes the problem of Byzantine-robust decentralized optimization. Section 3 presents the robust gossip framework as well as the main convergence guarantees. Then, Section 4 instantiates the general framework with several rules (and the associated robustness guarantees), and provides guarantees for a D-SGD algorithm based on F–RG. Finally, Section 5 presents the *Spectral Heterogeneity* attack, which is then used with other attacks to experimentally challenge several aggregation schemes in Section 6.

## 2. Background

### 2.1. Decentralized optimization.

We consider a system composed of $m$ computing units that communicate synchronously through a communication network, which is represented as an undirected graph $\mathcal{G}$. We denote by $\mathcal{H}$ the set of honest nodes, and $\mathcal{B}$ the (unknown) set of Byzantine nodes. Each unit $i$ holds a local parameter $\boldsymbol{x}_i \in \mathbb{R}^d$, a local loss function $f_i : \mathbb{R}^d \to \mathbb{R}$, and can communicate with its neighbors in the graph $\mathcal{G}$. We denote the set of neighbors of node $i$ by $n(i)$ and by $n_{\mathcal{H}}(i)$ (resp. $n_{\mathcal{B}}(i)$) the set of honest (resp. Byzantine) ones. We study decentralized algorithms for solving

$$\arg\min_{\boldsymbol{x} \in \mathbb{R}^d} \left\{ f_{\mathcal{H}}(\boldsymbol{x}) := \frac{1}{|\mathcal{H}|} \sum_{i \in \mathcal{H}} f_i(\boldsymbol{x}) \right\}. \qquad (1)$$

Due to the averaging nature of Equation (1), centralized algorithms for solving this problem rely on global averaging of the gradients computed at each node. In the decentralized setting, we rely on performing local (node-wise) inexact averaging steps instead.

**Gossip Communication.** Standard decentralized optimization algorithms typically rely on the so-called *gossip* communication protocol (Boyd et al., 2006; Nedic & Ozdaglar, 2009; Scaman et al., 2017; Kovalev et al., 2020). The gossip protocol consists of updating the parameter of a node $i$ through a linear combination of the parameters of its neighbors, with updates of the form $\boldsymbol{x}_i^{t+1} = \boldsymbol{x}_i^t - \eta \sum_{j=1}^m w_{ij}(\boldsymbol{x}_i^t - \boldsymbol{x}_j^t)$, where $w_{ij}$ is a weight associated with the edge $(ij)$ of the graph and $\eta \geq 0$ will denote a communication step-size. By denoting $\boldsymbol{W}$ the Laplacian matrix associated with the weighted graph $\mathcal{G}$, i.e $\boldsymbol{W}_{ij} = -w_{ij}$ if $i \neq j$ and $\boldsymbol{W}_{ii} = \sum_{j \in n(i)} w_{ij}$, and denoting the matrix of parameters as $\boldsymbol{X} = (\boldsymbol{x}_1, \dots, \boldsymbol{x}_m)^T$, then the gossip update conveniently writes:

$$\boldsymbol{X}^{t+1} = \boldsymbol{X}^t - \eta \boldsymbol{W} \boldsymbol{X}^t. \qquad (2)$$

The Laplacian matrix (a.k.a. gossip matrix), is symmetric non-negative. When the graph is connected, its kernel is restricted to the line of the constant vectors, i.e. $\mathrm{span}(1, \dots, 1)^T$. In the following, we will always consider that the graph $\mathcal{G}$ is weighted, so that a unique Laplacian matrix is associated with each graph $\mathcal{G}$. Moreover, we denote by $\mu_{\max}(\mathcal{G}_{\mathcal{H}})$ and $\mu_2(\mathcal{G}_{\mathcal{H}})$ the largest and smallest non-zero eigenvalues of the Laplacian matrix $\boldsymbol{W}_{\mathcal{H}}$ of the *honest subgraph* $\mathcal{G}_{\mathcal{H}}$, and by $\gamma = \mu_2(\mathcal{G}_{\mathcal{H}})/\mu_{\max}(\mathcal{G}_{\mathcal{H}})$ its *spectral gap*. Spectral properties of the Laplacian matrix are known to characterize the convergence of gossip optimization methods. For instance, in the absence of Byzantine nodes, Equation (2) with step-size $\eta \leq \mu_{\max}(\mathcal{G})^{-1}$ leads to a linear convergence of the nodes parameter values towards the average of the initial parameters: $\|\boldsymbol{X}^t - \overline{\boldsymbol{X}}^0\|^2 \leq$

$(1-\eta\mu_2(\mathcal{G}))^t\|\boldsymbol{X}^0-\overline{\boldsymbol{X}}^0\|^2$, for $\overline{\boldsymbol{X}}^0$ the matrix with columns $m^{-1}\sum_{j=1}^m \boldsymbol{x}_j^0$.

**Robustness Issue.** Gossip communication relies on updating the nodes' parameters by performing non-robust local averaging. As such, similarly to the centralized case, any Byzantine neighbor of node $i$ can drive the update to any desired value (Blanchard et al., 2017). Then, the poisoned information spreads through gossip communications.

## 2.2. Byzantine-robust decentralized optimization.

In this section, we describe the threat model and the robustness criterion that we consider.

**Threat model.** We consider Byzantine nodes to be unknown omniscient adversaries, able to collude and send distinct values to each of their neighbors. We measure their influence by considering the weight of Byzantine nodes in the neighborhood of each honest node, $(b(i) := \sum_{j\in n_\mathcal{B}(i)} w_{ij})_{i\in\mathcal{H}}$ (as in LeBlanc et al. (2013); He et al. (2023)), instead of the total number of Byzantine nodes $|\mathcal{B}|$ as done for the centralized or fully-connected setting. Similarly, we denote by $h(i) = \sum_{j\in n_\mathcal{H}(i)} w_{ij}$ the weights of the honest nodes adjacent to the node $i$.

In the case of an arbitrary communication network, the number of honest nodes does not provide enough information by itself, as the results depend on *how* the nodes are linked, i.e., the topology of the honest subgraph. Therefore, in the case of sparse topologies, it is necessary to consider a property of the graph related to its structure rather than relying solely on the number of honest nodes. We will show that spectral properties of the Laplacian of honest subgraph are relevant quantities for robustness analyses and introduce the following class of graphs.

**Definition 2.1.** For any $\mu_{\min} \geq 0$ and $b \geq 0$, we define the class of weighted graphs

$$\Gamma_{\mu_{\min},b} = \left\{ \mathcal{G} \text{ s.t. } \mu_2(\mathcal{G}_\mathcal{H}) \geq \mu_{\min} \text{ and } \max_{i\in\mathcal{H}} b(i) \leq b \right\}.$$

In other words, we introduce a subset of all possible graphs, partitioning in terms of (i) the second smallest eigenvalue of the honest subgraph, (a.k.a. the *algebraic connectivity*), that is restricted to be larger than a minimal value $\mu_{\min}$, and (ii) the maximal weight of Byzantines in the neighborhood of an honest node, which is restricted to be smaller than $b$.

Note that if all edges are equally weighted ($w_{ij} = \omega$), then $b(i) = |n_\mathcal{B}(i)| \cdot \omega$, and (ii) boils down to upper bounding the number of Byzantines in the neighborhood of honest nodes by $f := b \cdot \omega^{-1}$. Hence, Definition 2.1 is an extension of the standard *"there are at most $f$ byzantine nodes among the $|\mathcal{B}|+|\mathcal{H}|$ nodes"* to the setting of arbitrary connected graphs. We point out that this class of graphs depends on the spectral

properties *of the honest subgraph*, meaning that for a given communication network, these properties change depending on the location of Byzantine failures, and the associated graph can either fall within $\Gamma_{\mu_{\min},b}$ if Byzantines are "well spread", or not, e.g. if they are adversarially chosen.

**Approximate Average Consensus.** The average consensus problem consists in finding the average of vectors locally held by honest nodes $(\boldsymbol{x}_i)_{i\in\mathcal{H}}$. It is a specific case of Equation (1) obtained by considering $f_i(\boldsymbol{x}) = \|\boldsymbol{x} - \boldsymbol{x}_i\|^2$. Due to adversarial attacks only an *approximate* estimate of the average of honest nodes vector $\overline{\boldsymbol{x}}_\mathcal{H} := |\mathcal{H}|^{-1}\sum_{i\in\mathcal{H}} \boldsymbol{x}_i$ can be reached (Karimireddy et al., 2023). We therefore introduce the following criterion to assess the robustness of a communication algorithm.

**Definition 2.2** (*r*-robustness on $\mathcal{G}$.). Let $r < 1$. A communication algorithm Alg is *r*-robust on a graph $\mathcal{G}$ if from any initial local parameters $(\boldsymbol{x}_i)_{i\in\mathcal{H}} \in (\mathbb{R}^d)^\mathcal{H}$, it allows any honest node $i$ to compute a vector $\boldsymbol{x}_i^+$ such that

$$\frac{1}{|\mathcal{H}|}\sum_{i\in\mathcal{H}}\|\boldsymbol{x}_i^+ - \overline{\boldsymbol{x}}_\mathcal{H}\|^2 \leq r\frac{1}{|\mathcal{H}|}\sum_{i\in\mathcal{H}}\|\boldsymbol{x}_i - \overline{\boldsymbol{x}}_\mathcal{H}\|^2.$$

Imposing $r < 1$ means the honest nodes' parameters have to be strictly closer (on average) to the initial mean after the communication than before. It thus requires that the reduction of the *variance* of nodes parameters $\text{Var}_\mathcal{H}(\boldsymbol{x}) - \text{Var}_\mathcal{H}(\boldsymbol{x}^+)$ is larger than the *bias* $\|\overline{\boldsymbol{x}}_\mathcal{H}^+ - \overline{\boldsymbol{x}}_\mathcal{H}\|^2$ introduced by the Byzantines, with $\text{Var}_\mathcal{H}(\boldsymbol{x}) := |\mathcal{H}|^{-1}\sum_{i\in\mathcal{H}}\|\boldsymbol{x}_i - \overline{\boldsymbol{x}}_\mathcal{H}\|^2$. Remark that the *r*-robustness of an algorithm on a graph $\mathcal{G}$ states that a *single use* of the algorithm strictly reduces the average quadratic error. However, it does not mean that multiple uses would result in a geometric decrease; indeed, we cannot simply use induction as $\overline{\boldsymbol{x}}_\mathcal{H}^+ \neq \overline{\boldsymbol{x}}_\mathcal{H}$.

# 3. The Robust Gossip Framework

We now introduce a generic framework, which relies on two key building blocks: (i) a generic update form, depending on an aggregation function $F : (\mathbb{R}_+ \times \mathbb{R}^d)^n \to \mathbb{R}^d$, and (ii) a set of those aggregation functions F, referred to as *robust summation* functions. We then show that the combination of these two blocks, *i.e.*, the generic update used with a robust summation function F, leads to a robust decentralized algorithm, hence the name: Robust Gossip.

We finally show that this framework leads to (near-)optimal robustness guarantees for well-chosen F.

## 3.1. The Robust Gossip Method: adding robust differences instead of averaging robustly.

We call *aggregation rule* a function $F : (\mathbb{R}_+ \times \mathbb{R}^d)^n \to \mathbb{R}^d$, meant to aggregate a set of vectors $(z_i)_{i\in[n]} \in (\mathbb{R}^d)^n$ with weights $(w_i)_{i\in[n]} \in \mathbb{R}_+^n$. For $\eta > 0$ a communication step-size, the associated robust gossip algorithm (coined F–RG)

consists, for any honest node $i \in \mathcal{H}$ of the communication network $\mathcal{G}$, in updating its parameter $\boldsymbol{x}_i$ using:

$$\boldsymbol{x}_i^+ := \boldsymbol{x}_i - \eta F\left((w_{ij}, \boldsymbol{x}_i - \boldsymbol{x}_j)_{j \in n(i)}\right). \qquad \text{(F–RG)}$$

Crucially, each node applies the (robust) aggregation rule F to $(\omega_j, \boldsymbol{z}_j)_{j \in n(i)} := (w_{ij}, \boldsymbol{x}_i - \boldsymbol{x}_j)_{j \in n(i)}$, i.e. to *the differences of its parameter with those of its neighbors*, and uses this estimate to update its parameter. Thus, Equation (F–RG) recovers the standard gossip update from Equation (2) if F is the weighted sum operator (which is unfortunately not robust). Hence, we will look for aggregation functions that are robust versions of the weighted sum.

In contrast, directly averaging *the parameters* of the neighbors, even with a robust aggregation rule, would highly suffer from heterogeneity. Indeed, this leads to a *biased estimate* of the mean of initial parameters for sparse communication graphs. Extra assumptions, such as homogeneity of the local objectives, are thus needed to alleviate this problem (Fang et al., 2022).

Meanwhile, the RG method uses intrinsically decentralized updates, allowing for tight convergence guarantees in sparse communication graphs with heterogeneous local objectives. The strength of the RG framework is to turn any robust summation into a robust gossip algorithm. As such, one can focus on the design of robust aggregation functions without worrying about the decentralized aspect.

### 3.2. Robust Summation Functions.

Our analysis of F–RG relies on aggregation rules that meet the following robustness conditions.

**Definition 3.1** $((b, \rho)$–robust summation). Let $b, \rho \geq 0$. An aggregation rule $F : (\mathbb{R}_+ \times \mathbb{R}^d)^n \to \mathbb{R}^d$ is a $(b, \rho)$–*robust summation* if, for any vectors $(\boldsymbol{z}_i)_{i \in [n]} \in (\mathbb{R}^d)^n$, any weights $(\omega_i)_{i \in [n]} \in \mathbb{R}_+^n$ and any $S \subset [n]$ such that $\sum_{i \in \overline{S}} \omega_i \leq b$ (where $\overline{S} := [n] \backslash S$),

$$\left\| F\left((\omega_i, \boldsymbol{z}_i)_{i \in [n]}\right) - \sum_{i \in S} \omega_i \boldsymbol{z}_i \right\|^2 \leq \rho b \sum_{i \in S} \omega_i \|\boldsymbol{z}_i\|^2.$$

In (F–RG), $S$ is the set of honest neighbors $n_{\mathcal{H}}(i)$, while $\overline{S}$ is the set of Byzantine neighbors $n_{\mathcal{B}}(i)$. We will exhibit several $(b, \rho)$–robust summation rules, including a practical rule for $\rho = 2$, in Section 4.

*Remark* 3.2. The latter definition differs from $(f, \kappa)$–robustness (Allouah et al., 2023) we upper bound the error using the *second moment* of vectors within $S$ instead of their *variance*. Note also that $(f, \kappa)$–robustness is stated with constant weights only. Therefore, if $F$ is $(f, \kappa)$–robust, then $F$ is a $(b, \rho)$-robust summation with e.g. uniform weights $\omega_i = 1/(n - f)$ with $b = f/(n - f)$ and $\rho = \kappa/b$.

### 3.3. Convergence of RG under $(b, \rho)$-robustness.

As briefly discussed in the introduction, the goal of communicating is to reduce the variance, which comes at the price of bias. This is unavoidable since communicating allows nodes to inject wrong information which biases the system.

In the following core result, we show how $(b, \rho)$–robustness enables us to tightly quantify how much a single step of F–RG reduces the variance, and how much bias is injected. Recall that $\text{Var}_{\mathcal{H}}(\boldsymbol{x}) = \frac{1}{|\mathcal{H}|} \sum_{i \in \mathcal{H}} \|\boldsymbol{x}_i - \overline{\boldsymbol{x}}_{\mathcal{H}}\|^2$ is the variance of honest nodes.

**Theorem 3.3.** *Let $F$ be a $(b, \rho)$–robust summation, $b$ and $\mu_{\min}$ be s.t. $2\rho b \leq \mu_{\min}$, and $\mathcal{G} \in \Gamma_{\mu_{\min}, b}$. Then, assuming $\eta \leq \mu_{\max}(\mathcal{G}_{\mathcal{H}})^{-1}$, the output $(\boldsymbol{x}_i^+)_{i \in \mathcal{H}}$ of F–RG verifies:*

$$\begin{cases} \frac{1}{|\mathcal{H}|} \sum_{i \in \mathcal{H}} \|\boldsymbol{x}_i^+ - \overline{\boldsymbol{x}}_{\mathcal{H}}\|^2 \leq (1 - \eta\,(\mu_{\min} - 2\rho b))\,\text{Var}_{\mathcal{H}}(\boldsymbol{x}), \\ \|\overline{\boldsymbol{x}}_{\mathcal{H}}^+ - \overline{\boldsymbol{x}}_{\mathcal{H}}\|^2 \leq 2\rho b\,\eta\,\text{Var}_{\mathcal{H}}(\boldsymbol{x}). \end{cases}$$

*Thus, F–RG is $(1 - \eta\,(\mu_{\min} - 2\rho b))$–robust for $\mathcal{G} \in \Gamma_{\mu_{\min}, b}$.*

While the bound on $\eta$ depends on the honest subgraph, as $\mu_{\max}(\mathcal{G}_{\mathcal{H}}) \leq \mu_{\max}(\mathcal{G})$, $\eta$ can be set conservatively by evaluating $\mu_{\max}$ on the whole graph. Note that while $r$–robustness is guaranteed for the whole class $\Gamma_{\mu_{\min}, b}$, the value of $r$ will depend on the actual graph within the class.

**Chaining aggregation steps.** When low variance levels are required, it is necessary to perform several robust gossip steps one after the other. This contrasts with the centralized setting, in which the variance can be brought to zero in one step. While the variance reduces at a linear rate, the bias accumulates as multiple aggregation steps are performed. We provide bounds for $t$ steps of F–RG in the following Corollary.

**Corollary 3.4.** *Let $F$ be a $(b, \rho)$–robust summation, let $b$ and $\mu_{\min}$ be such that $2\rho b \leq \mu_{\min}$, and let $\mathcal{G} \in \Gamma_{\mu_{\min}, b}$. We denote $\delta = \frac{2\rho b}{\mu_{\min}}$ and $\gamma = \mu_{\min}/\mu_{\max}(\mathcal{G}_{\mathcal{H}})$. Then, for $(\boldsymbol{x}_i^t)_{i \in \mathcal{G},\, t \geq 0}$ obtained from any $(\boldsymbol{x}_i^0)_{i \in \mathcal{G}}$ through $(\boldsymbol{x}_i^{t+1})_{i \in \mathcal{G}} = \text{F–RG}((\boldsymbol{x}^t)_{i \in \mathcal{G}})$, with $\eta = \mu_{\max}(\mathcal{G}_{\mathcal{H}})^{-1}$,*

$$\begin{cases} \text{Var}_{\mathcal{H}}(\boldsymbol{x}^t) \leq (1 - \gamma(1 - \delta))^t \, \text{Var}_{\mathcal{H}}(\boldsymbol{x}^0), \\ \|\overline{\boldsymbol{x}}_{\mathcal{H}}^t - \overline{\boldsymbol{x}}_{\mathcal{H}}^0\| \leq \frac{\sqrt{\gamma\delta}\left(1 - [1 - \gamma(1-\delta)]^{t/2}\right)}{1 - \sqrt{1 - \gamma(1-\delta)}}\sqrt{\text{Var}_{\mathcal{H}}(\boldsymbol{x}^0)}. \end{cases}$$

*Consensus is thus reached, as $\text{Var}_{\mathcal{H}}(\boldsymbol{x}^t) \to_{t \to \infty} 0$, and*

$$\|\overline{\boldsymbol{x}}_{\mathcal{H}}^t - \overline{\boldsymbol{x}}_{\mathcal{H}}^0\|^2 \leq \frac{4\delta}{\gamma(1 - \delta)^2}\text{Var}_{\mathcal{H}}(\boldsymbol{x}^0). \qquad (3)$$

While the total L2 error (bias plus variance) is guaranteed to decrease after a single step by Theorem 3.3, it may increase if several F–RG steps are performed because of bias accumulation. This happens when the factor multiplying the

variance in Equation (3) is larger than 1, which essentially means $\gamma \leq \delta$. Despite this bias, the output of the resulting robust aggregation procedure is (arbitrarily) close to consensus, which can be desirable. Proofs of Theorem 3.3 and Corollary 3.4 are respectively given in Appendices D.1 and D.2.

**Dependence on the parameters.** As expected, the bias increases with the amount of Byzantine corruption (through $\delta$) and decreases as the graph becomes more connected (i.e, $\gamma \to 1$). One can then use parameter $\eta$ (up to its maximum value) to control the bias-variance trade-off.

### 3.4. Spectral limit of r-robust decentralized algorithms.

We now provide an *upper bound* on the weight of Byzantine neighbors that can be tolerated by any algorithm running on a communication network in which the honest subgraph has a given algebraic connectivity.

**Theorem 3.5.** *Let $\mu_{\min} \geq 0$, $b \geq 0$ be such that $\mu_{\min} \leq 2b$. Then for any $h \geq 0$ and any algorithm* Alg, *there exists a graph $\mathcal{G} \in \Gamma_{\mu_{\min},b}$ in which all honest nodes have a weight of honest neighbors $h(i)$ larger than $h$, and such that for any $r < 1$,* Alg *is not $r$–robust on $\mathcal{G}$.*

We refer the reader to Appendix E for the proof details. It follows from Theorem 3.5 that when a theoretical guarantee quantifies the robustness of an algorithm on a graph through $\mu_{\min}$, we must have $2b < \mu_{\min}$. Importantly, this upper bound on the breakdown point is *independent of the total weight of honest neighbors*.

For a fully-connected graph with uniform weights $\omega$, we have $\mu_2(\mathcal{G}_{\mathcal{H}}) = \omega|\mathcal{H}|$ and $b(i) = \omega|\mathcal{B}|$. Then, the previous condition boils down to $|\mathcal{H}| > 2|\mathcal{B}|$, i.e. there is less than $1/3$ of Byzantine nodes, which recovers known necessary robustness conditions of distributed system (Lamport et al., 1982; Vaidya et al., 2012; El-Mhamdi et al., 2021).

**Near-optimal breakdown point.** Theorem 3.5 states that uniformly ensuring $r$–robustness on $\Gamma_{\mu_{\min},b}$ is impossible as soon as $2b \geq \mu_{\min}$, and we know that update (F–RG) is $r$-robust as soon as $2\rho b < \mu_{\min}$. Therefore, no $(\rho, b)$-robust summand exists with $\rho < 1$, and (F–RG) achieves the optimal breakdown if a $(b, 1)$-robust aggregation rule is used. Such rules exist, as shown in Section 4, but are unfortunately not practical, as they require prior knowledge on the Byzantine nodes. It is an open question whether $\rho = 1$ can be achieved using a practical rule.

Nevertheless, we propose a practical robust summation rule that achieves $\rho = 2$, thus robust if $4b < \mu_{\min}$. This is a significant improvement over existing works. For example, in He et al. (2023), the $4b < \mu_{\min}$ condition is essentially replaced by $cb \leq \gamma\mu_{\min}$ (*e.g.*, for regular graphs), where $c > 0$ is a large constant, and $\gamma = \mu_2(\mathcal{G}_{\mathcal{H}})/\mu_{\max}(\mathcal{G}_{\mathcal{H}})$ the

graph's spectral gap. This gap rapidly shrinks with the size and the lack of connectivity of the graph, making the condition orders of magnitude worse for large sparse graph. In Wu et al. (2023), the breakdown condition is $8b\sqrt{|\mathcal{H}|} \leq \mu_{\min}$, which means that the robustness guarantee decreases when the number of honest nodes increases. For instance, only a $1/(9|\mathcal{H}|^{1/2})$ fraction of Byzantine nodes is tolerated for a fully-connected network, whereas we tolerate up to $1/5$.

We conclude this section by two remarks on potential alternative characterizations of the breakdown point.

*Remark* 3.6 (On algebraic connectivity). Theorem 3.5 does not imply that a given communication algorithm systematically fails as soon as $2b \geq \mu_2(\mathcal{G})$, but rather that since there exists a graph for which it is the case, one can not have an $r$–robust algorithm with an assumption based on $\mu_2(\mathcal{G})$ and $b$ looser than $\mu_2(\mathcal{G}) \geq 2b$. Yet, one can still prove breakdown points using other graph-related quantities, which might lead to tolerating $b > \mu_{\min}/2$ Byzantine nodes for specific graph architectures. This gap is standard in the decentralized optimization literature, where optimal algorithms depend on the (square root of the) *spectral gap* of the gossip matrix (Scaman et al., 2017; Kovalev et al., 2020), whereas iteration lower bounds are proven in terms of diameter of the communication graph.

*Remark* 3.7 (Dimension-dependent breakdown points). The Approximate Average Consensus problem (Section 2.2) is related to the *Approximate Consensus Problem* (ACP) (Dolev et al., 1986), in which the nodes must converge to the same value while *remaining within the convex hull* of the initial parameters. This is a harder problem, since the ACP cannot be solved using iterative communication on a system of $m$ nodes with $f$ Byzantine failures in dimension $d$ if $m \leq (d + 2)f + 1$ (Vaidya, 2014). This dependence on the dimension $d$ is prohibitive for ML applications. On the contrary, our definition of $r$–robustness only requires the algorithm to improve the average squared distance to the target value, which is enough for D-SGD to converge, and enables us to prove *dimension-independent* breakdown. Yet, it would be interesting to link their notion of $r$-robust networks (LeBlanc et al., 2013) with algebraic connectivity.

## 4. From the general framework to practical decentralized algorithms

In this section, we first define robust summation rules and link our general framework with existing decentralized robust algorithms in Section 4.1. Then we prove convergence for Decentralized-SGD based on F–RG, in Section 4.2.

### 4.1. Examples of $(b, \rho)$-robust rules

Several methods have been proved to be $(f, \kappa)$-robust, including the Coordinate-Wise Trimmed Mean (CWTM) (Yin

et al., 2018), the Coordinate-wise Median (CWM) (Yin et al., 2018), the Geometric Median (GM) (Yin et al., 2018; Pillutla et al., 2022) and Krum (Blanchard et al., 2017). It follows from Remark 3.2 that they are also $(b, \rho)$–robust summation.

However, since $(b, \rho)$–robust summation is weaker than $(f, \kappa)$–robustness, we can introduce robust aggregation methods using this new perspective with tighter robustness guarantees. The following introduced aggregator either recover existing algorithm, or are tighter than previous approach. In their definition, $(\omega_i, \boldsymbol{z}_i)_{i\in[n]} \in (\mathbb{R}_+ \times \mathbb{R}^d)^n$ and $S \subset [n]$ such that $\sum_{\overline{S}} \omega_i \leq b$.

**Geometric Trimmed Sum (GTS).** Assume w.l.o.g. that $(\|\boldsymbol{z}_i\|)_{i\in[n]}$ are sorted, i.e. $\|\boldsymbol{z}_1\| \leq \ldots \leq \|\boldsymbol{z}_n\|$, and we denote as $k^*(b) := \max\{k \in [n]; \sum_{i\geq k} \omega_i \geq b\}$ the index of the largest vector which has at least a weight $b$ of vectors larger than it[1]. (GTS) computes $\tilde{\omega}_{k^*(b)} := \sum_{i\geq k^*(b)} \omega_i - b$, and outputs

$$\text{GTS}\big((\omega_i, \boldsymbol{z}_i)_{i\in[n]}\big) = \tilde{\omega}_{k^*(b)}\boldsymbol{z}_{k^*} + \sum_{i<k^*(b)} \omega_i \boldsymbol{z}_i.$$

In the simpler case where the weights are 1 and $b \in [n]$, GTS consists in discarding the $b$ largest vectors within $(\boldsymbol{z}_i)_{i\in[n]}$ and summing the rest.

**Clipped Sum (CS).** Given a threshold function $\tau : (\mathbb{R}_+ \times \mathbb{R}^d)^n \mapsto \mathbb{R}_+$, output the mean of clipped vectors:

$$\text{CS}\big((\omega_i, \boldsymbol{z}_i)_{i\in[n]}; \tau\big) := \frac{1}{n}\sum_{i=1}^n \omega_i \, \text{Clip}\big(\boldsymbol{z}_i; \tau\big((\omega_i, \boldsymbol{z}_i)_{i\in[n]}\big)\big),$$

where $\forall \boldsymbol{z} \in \mathbb{R}^d, \tilde{\tau} \in \mathbb{R}_+, \text{Clip}(\boldsymbol{z}; \tilde{\tau}) := \frac{\boldsymbol{z}}{\|\boldsymbol{z}\|}\min(\|\boldsymbol{z}\|, \tilde{\tau})$.

We propose the following threshold function[2], which leads to a practical and nearly optimal aggregator.

**Practical Clipping (CS₊).** Let $\text{CS}_+ := \text{CS}\big(\cdot; \tau_+\big)$, where

$$\tau_+\big((\omega_i, \boldsymbol{z}_i)_{i\in[n]}\big) := \max\left\{\tau \geq 0 : \sum_{i=1}^n \omega_i \mathbf{1}_{\|\boldsymbol{z}_i\|\geq\tau} \geq 2b\right\}.$$

This rule corresponds, in the specific case of unitary weights $\omega_i = 1$ and $b \in [n]$, to defining the clipping threshold as the $2b^{th}$ largest value within $\|\boldsymbol{z}_1\|, \ldots, \|\boldsymbol{z}_n\|$ (i.e., $\|\boldsymbol{z}_{k^*(2b)}\|$). We now provide a robustness guarantee for these two aggregation rules in the following theorem.

**Theorem 4.1.** *Let $b \geq 0$, then GTS is $(b, \rho)$–robust with $\rho = 4$, and $\text{CS}_+$ is $(b, \rho)$–robust with $\rho = 2$.*

---

[1]When weights sum to 1, $k^*$ is a quantile function.

[2]Note that Clipped Sum cannot be a $(b, \rho)$–robust summation when the threshold function is a fixed constant $\tau \geq 0$: the clipping threshold must be adaptive to the input vectors.

*Remark* 4.2 (Concurrent work). Allouah et al. (2025) study the influence of an adaptive clipping scheme, named Adaptive Robust Clipping (ARC), with the same adaptive clipping threshold as $\text{CS}_+$. However, they use it *before* any aggregation function $(f, \kappa)$–robust $F$, and only analyze the robustness of $\text{F} \circ \text{ARC}$, making their approach orthogonal to ours. Moreover, their focus is on the federated case.

Next, we define *oracle* (or.) threshold functions. Those require the knowledge of the set $S$ to be computed, which eventually corresponds to being able to identify which node is honest and which node is Byzantine. This is obviously not a reasonable assumption in practice. Even though, it shows that the optimality gap is not inherent to clipping, since an oracle choice of threshold is optimal. Furthermore, the guarantees from He et al. (2023) rely on such assumptions[3].

**Oracle Clipping.** Let $\text{CS}_+^{\text{or.}} := \text{CS}\big(\cdot; \tau_+^{\text{or.}}\big)$, where $\tau_+^{\text{or.}}\big((\omega_i, \boldsymbol{z}_i)_{i\in[n]}; S\big) = \max\{\tau \geq 0 : \sum_{i\in S} \omega_i \mathbf{1}_{\|\boldsymbol{z}_i\|\geq\tau} \geq b\}$.

**Oracle clipping (He et al., 2023).** $\text{CS}_{\text{He}}^{\text{or.}} := \text{CS}\big(\cdot; \tau_{\text{He}}^{\text{or.}}\big)$, where $\tau_{\text{He}}^{\text{or.}}\big((\omega_i, \boldsymbol{z}_i)_{i\in[n]}; S\big) = \sqrt{\frac{1}{b}\sum_{i\in S}\omega_i\|\boldsymbol{z}_i\|^2}$.

As shown below, $\text{CS}_+^{\text{or.}}$ leads to an optimal breakdown point. On the contrary, $\text{CS}_{\text{He}}^{\text{or.}}$ only achieves $\rho = 4$, despite its oracle aspect. Proofs of Theorems 4.1 and 4.3 are given in Appendix F.

**Theorem 4.3.** *Let $b \geq 0$, then $\text{CS}_+^{or.}$ is $(b, \rho)$–robust with $\rho = 1$, and $\text{CS}_{He}^{or.}$ is $(b, \rho)$–robust with $\rho = 4$.*

When instantiated in specific settings, our framework gives tight convergence guarantees (improving on the existing ones) for existing algorithms.

**Proposition 4.4.** (F–RG) *recovers existing algorithms.*

1. GTS–RG, *on a fully connected communication network $\mathcal{G}$ with constant weight, corresponds to Nearest Neighbors Averaging (Farhadkhani et al., 2022, NNA).*
2. $\text{CS}_{He}^{or.}$–RG *recovers* ClippedGossip *(He et al., 2023).*

### 4.2. Byzantine robust Distributed SGD on graphs

We now give convergence results for a D-SGD-type algorithm that uses (F–RG) for robust decentralized aggregation. Several works on Byzantine-robust SGD abstract away the aggregation procedure by relying on contraction properties (Karimireddy et al., 2021; Wu et al., 2023; Farhadkhani et al., 2023), so that global D-SGD convergence follows from the robustness of the averaging procedure. Our Corollary 4.8 builds on the $(\alpha, \lambda)$-reduction from Farhadkhani et al. (2023):

**Definition 4.5** ($(\alpha, \lambda)$-reduction)**.** A coordinating phase $\Psi$ verifies an $(\alpha, \lambda)$-reduction if, from any initial local param-

---

[3]In their experiments, they propose a practical (i.e. non-oracle) threshold function that is not supported by theory.

eters $(\boldsymbol{x}_i)_{i \in \mathcal{H}} \in (\mathbb{R}^d)^{\mathcal{H}}$, each honest nodes $i \in \mathcal{H}$ obtain a parameter vector $\boldsymbol{x}_i^+$ upon the completion of $\Psi$ such that

$$\frac{1}{|\mathcal{H}|} \sum_{i \in \mathcal{H}} \|\boldsymbol{x}_i^+ - \overline{\boldsymbol{x}}_{\mathcal{H}}^+\|^2 \leq \alpha \frac{1}{|\mathcal{H}|} \sum_{i \in \mathcal{H}} \|\boldsymbol{x}_i - \overline{\boldsymbol{x}}_{\mathcal{H}}\|^2,$$

$$\|\overline{\boldsymbol{x}}_{\mathcal{H}}^+ - \overline{\boldsymbol{x}}_{\mathcal{H}}\|^2 \leq \lambda \frac{1}{|\mathcal{H}|} \sum_{i \in \mathcal{H}} \|\boldsymbol{x}_i - \overline{\boldsymbol{x}}_{\mathcal{H}}\|^2.$$

*Remark* 4.6. $(\alpha, \lambda)$-reduction and $r$-robustness are closely related quantities since $(\alpha, \lambda)$ reductions implies $r$-robustness with $r \leq \alpha + \lambda$, and $r$-robustness implies $\alpha, \lambda \leq r$. Yet, $r$-robustness explicitly requires that $r < 1$, unlike $(\alpha, \lambda)$-reduction. In essence, $r$-robustness expresses more precisely that *nodes benefit from the communication*.

The $(\alpha, \lambda)$ requirements on the aggregation procedure exactly match the guarantees of Theorem 3.3: using one single step of F–RG as a coordination phase leads to $\alpha = 1 - \gamma(1 - \delta)$ and $\lambda = \gamma\delta$, and using multiple communication steps leads to $\alpha \approx 0$ and $\lambda = {}^{4\delta}/_{\gamma(1-\delta)^2}$. We build on this abstraction to propose a Byzantine robust decentralized stochastic gradient descent framework.

We consider Problem 1, where we assume that each local function $f_i$ is a risk computed using a loss $\ell$ on a data distribution $\mathcal{D}_i$, i.e $f_i(\boldsymbol{x}) = \mathbb{E}_{\boldsymbol{\xi} \sim \mathcal{D}_i}[\nabla \ell(\boldsymbol{x}, \boldsymbol{\xi})]$. We solve Problem 1 using D-SGD over a communication network $\mathcal{G}$. Robustness to Byzantine nodes is obtained using (F–RG) as the aggregation rule, coupled with Polyak momentum used as a moving average to reduce the stochastic noise.

---

**Algorithm 1** Byzantine-Resilient D-SGD with F–RG

---

**Input:** Initial model $\boldsymbol{x}_i^0 \in \mathbb{R}^d$, local loss functions $f_i$, initial momentum $m_i^0 = 0$, momentum coefficient $\beta = 0$, learning rate $\eta_{op}$, communication step size $\eta$, communication graph $\mathcal{G}$, upper bound on Byzantine weight $b$.
**for** $t = 0$ **to** $T$ **do**
    **for** $i \in \mathcal{H}$ **in parallel do**
        Sample a noisy gradient: $\boldsymbol{g}_i^t = \nabla f_i(\boldsymbol{x}_i^t) + \boldsymbol{\xi}_i^t$.
        Update the momentum: $\boldsymbol{m}_i^t = \beta \boldsymbol{m}_i^{t-1} + (1 - \beta)\boldsymbol{g}_i^t$.
        Optimization step: $\boldsymbol{x}_i^{t+1/2} = \boldsymbol{x}_i^t - \eta_{op}\boldsymbol{m}_i^t$.
        Send $\boldsymbol{x}_i^{t+1/2}$ to the neighbors $n(i)$.
        Update the model with F–RG
        $\boldsymbol{x}_i^{t+1} = \text{F-RG}\left(\boldsymbol{x}_i^{t+1/2}; \{\boldsymbol{x}_j^{t+1/2}; \ j \in n(i)\}\right).$
    **end for**
**end for**

---

The convergence results of this algorithm rely on the following standard assumptions.

**Assumption 4.7.** Objective functions regularity.

1. **(Smoothness)** There exists $L \geq 0$, s.t. $\forall \boldsymbol{x}, \boldsymbol{y} \in \mathbb{R}^d$, $\|\nabla f_i(\boldsymbol{x}) - \nabla f_i(\boldsymbol{y})\| \leq L\|\boldsymbol{x} - \boldsymbol{y}\|$.

2. **(Bounded noise)** There exists $\sigma \geq 0$ s.t. $\forall \boldsymbol{x} \in \mathbb{R}^d$, $\mathbb{E}[\|\nabla \ell(\boldsymbol{x}, \xi_i) - \nabla f_i(\boldsymbol{x})\|^2] \leq \sigma^2$, for all $i \in [\mathcal{H}]$.

3. **(Heterogeneity)** There exist $\zeta \geq 0$ s.t. $\forall \boldsymbol{x} \in \mathbb{R}^d$, $\frac{1}{\mathcal{H}} \sum_{i \in \mathcal{H}} \|\nabla f_i(\boldsymbol{x}) - \nabla f_{\mathcal{H}}(\boldsymbol{x})\|^2 \leq \zeta^2$.

We can now state the guarantees of Algorithm 1.

**Corollary 4.8.** *Let $F$ a $(b, \rho)$–robust summation, let $b \geq 0$ and let $\mathcal{G}$ a weighted graph such that $\delta = 2\rho/\mu_2(\mathcal{G}_{\mathcal{H}}) < 1$ and of spectral gap $\gamma = \mu_2(\mathcal{G}_{\mathcal{H}})/\mu_{\max}(\mathcal{G}_{\mathcal{H}})$. Under Assumption 4.7, for all $i \in \mathcal{H}$, the iterates produced by Algorithm 1 on $\mathcal{G}$ with $\eta = 1/\mu_{\max}(\mathcal{G})$ and learning rate $\eta_{op} = \mathcal{O}(1/\sqrt{T})$ (depending also on problem parameters such as $L$, $\gamma$ or $\delta$), verify as $T$ increases:*

$$\frac{1}{T}\sum_{t=1}^{T}\mathbb{E}\left[\|\nabla f_{\mathcal{H}}(\boldsymbol{x}_i^t)\|^2\right] = \mathcal{O}\left(\frac{L\sigma}{\gamma(1-\delta)\sqrt{T}} + \frac{\zeta^2}{\gamma^2(1-\delta)^2}\right)$$

$$\text{Var}_{\mathcal{H}}(\boldsymbol{x}^T) = \mathcal{O}\left(\frac{1}{T}\left(1 + \frac{\zeta^2}{\sigma^2}\right)\right).$$

*Performing $\tilde{\mathcal{O}}(\gamma^{-1}(1 - \delta)^{-1})$ steps of F–RG between each gradient computation leads to:*

$$\frac{1}{T}\sum_{t=1}^{T}\mathbb{E}\left[\|\nabla f_{\mathcal{H}}(\boldsymbol{x}_i^t)\|^2\right] = \mathcal{O}\left(\frac{L\sigma}{\sqrt{T}}\sqrt{\frac{\delta}{\gamma(1-\delta)^2}} + \frac{\delta\zeta^2}{\gamma(1-\delta)}\right).$$

It follows that those guarantees state that performing more aggregation steps between gradient computations improves the asymptotic error but under an additional communication cost.

This corollary is the consequence of the combination of our Theorem 3.3 with Theorem 1 of Farhadkhani et al. (2023). The result is simplified using $\delta \geq |\mathcal{H}|^{-1}$, and $\gamma \ll 1$. We refer the reader to Appendix G for a more precise result and a detailed proof.

## 5. Attacking robust gossip algorithms

In this section, we design an attack that aims to disrupt robust gossip algorithms. To do this, we model communications as perturbations of a gossip scheme, and analyze their impact on the variance among nodes, which allows us to deduce what perturbation Byzantines nodes should enforce for effective attacks. Recall that $\boldsymbol{X}_{\mathcal{H}}^t = (\boldsymbol{x}_1^t, \ldots, \boldsymbol{x}_{|\mathcal{H}|}^t)^T \in \mathbb{R}^{|\mathcal{H}| \times d}$ denotes the matrix of honest parameters at communication round $t$. Each step of F–RG can be decomposed as a perturbed gossip update (cf. Lemma D.1),

$$\boldsymbol{X}_{\mathcal{H}}^{t+1} = (\boldsymbol{I}_{\mathcal{H}} - \eta\boldsymbol{W}_{\mathcal{H}})\boldsymbol{X}_{\mathcal{H}}^t + \eta\boldsymbol{E}^t. \tag{4}$$

Where $\boldsymbol{E}^t$ is the perturbation term due to Byzantine nodes. In the following, we assume that $[\boldsymbol{E}^t]_i = \zeta_i^t \boldsymbol{a}_i^t$ for any honest node $i$, where $\boldsymbol{a}_i^t$ is the direction of attack on node

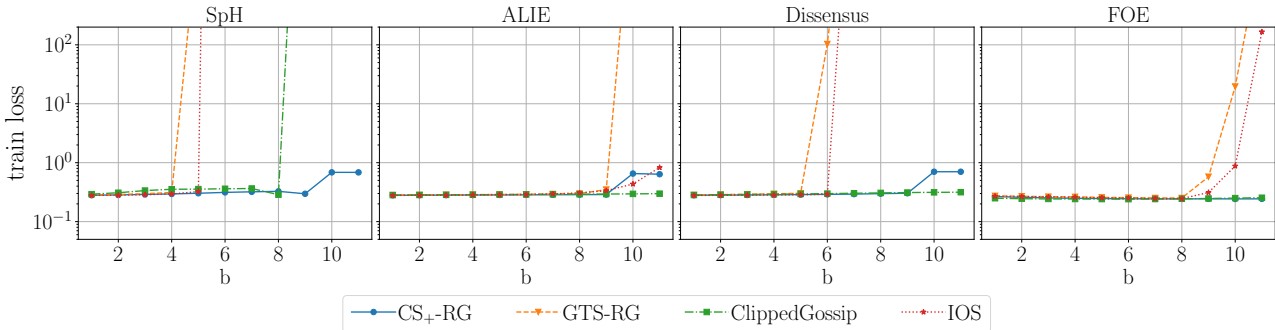

*Figure 1.* Training loss achieved by GTS–RG, CS$_+$–RG, ClippedGossip and IOS on MNIST ($\alpha = 1$) after 300 optimization and communication steps. The honest subgraph graph is $\mathcal{G}_\mathcal{H} := [\mathcal{G}_{m=13,k=8,c=1}]_\mathcal{H}$, as defined in Appendix E. Thus, $\mu_2(\mathcal{G}_\mathcal{H}) = 16$.

$i$, and $\zeta_i^t$ is a scaling factor, which is chosen to bypass the defenses. Typically, if $\zeta_i^t$ is small, a Byzantine node $j \in n_\mathcal{B}(i)$ can declare to node $i$ the parameter $\boldsymbol{x}_j^t = \boldsymbol{x}_i^t + \zeta_i^t \boldsymbol{a}_i^t$.

**Dissensus Attack.** Byzantine nodes can disrupt decentralized communication by maximizing the variance of the honest parameters. A natural decentralized notion of variance is the *Laplacian heterogeneity* $\sum_{i,j \in \mathcal{H}} w_{ij} \|\boldsymbol{x}_i - \boldsymbol{x}_j\|^2$, which corresponds to $\|\boldsymbol{X}_\mathcal{H}\|_{\boldsymbol{W}_\mathcal{H}}^2$. Finding $\boldsymbol{a}_i^t$ such that this heterogeneity is maximized at $t+1$ writes

$$\underset{[\boldsymbol{E}^t]_i = \zeta_i^t \boldsymbol{a}_i^t}{\arg\max} \|(\boldsymbol{I}_\mathcal{H} - \eta\boldsymbol{W}_\mathcal{H})\boldsymbol{X}_\mathcal{H}^t + \eta\boldsymbol{E}^t\|_{\boldsymbol{W}_\mathcal{H}}^2$$
$$= \underset{[\boldsymbol{E}^t]_i = \zeta_i^t \boldsymbol{a}_i^t}{\arg\max} \, 2\eta\langle\boldsymbol{W}_\mathcal{H}\boldsymbol{X}_\mathcal{H}^t, \boldsymbol{E}^t\rangle + o(\eta^2).$$

For small $\eta$, this suggests to take $\boldsymbol{a}_i^t = [\boldsymbol{W}_\mathcal{H}^t\boldsymbol{X}_\mathcal{H}^t]_i = \sum_{j \in n_\mathcal{H}(i)} w_{ij}(\boldsymbol{x}_i^t - \boldsymbol{x}_j^t)$. This choice of $\boldsymbol{a}_i^t$ corresponds to the *Dissensus* attack proposed in He et al. (2023). However, as gossip communication is usually operated with multiple communication rounds, maximizing only the pairwise differences at the next step is a short-sighted approach.

**Spectral Heterogeneity Attack.** Byzantine nodes can take into account the fact that several rounds of communication will occur, and focus on increasing the heterogeneity over the long term. This leads, at any time $t$, to maximizing for any $s \geq 0$ the pairwise differences at time $t + s$, i.e, finding

$$\underset{[\boldsymbol{E}^t]_i = \zeta_i^t \boldsymbol{a}_i^t}{\arg\max} \, 2\eta\langle\boldsymbol{W}_\mathcal{H}(\boldsymbol{I}_\mathcal{H} - \eta\boldsymbol{W}_\mathcal{H})^{2s+1}\boldsymbol{X}_\mathcal{H}^t, \boldsymbol{E}^t\rangle + o(\eta^2).$$

Taking $s \to +\infty$ leads to approximating $\boldsymbol{W}_\mathcal{H}(\boldsymbol{I}_\mathcal{H} - \eta\boldsymbol{W}_\mathcal{H})^{2s}$ as a projection on its eigenspace associated with the largest eigenvalue of $\boldsymbol{W}_\mathcal{H}(\boldsymbol{I}_\mathcal{H} - \eta\boldsymbol{W}_\mathcal{H})^{2s}$. This eigenspace corresponds to the space spanned by the eigenvector of $\boldsymbol{W}_\mathcal{H}$ associated with the smallest non-zero eigenvalue of $\boldsymbol{W}_\mathcal{H}$, i.e $\mu_2(\mathcal{G}_\mathcal{H})$. This eigenvector (denoted $\boldsymbol{e}_{fied}$) is commonly referred to as the *Fiedler vector* of the graph. Its coordinates essentially sort the nodes of the graph with the two farthest nodes associated with the largest and smallest value. Hence, the signs of the values in the Fiedler vector are typically used to partition the graph into two

(least-connected) components. Our *Spectral Heterogeneity* attack consists in taking $\boldsymbol{a}_i^t = [\boldsymbol{e}_{fied}\boldsymbol{e}_{fied}^T\boldsymbol{X}_\mathcal{H}^t]_i$, which leads Byzantine nodes to cut the graph into two by pushing honest nodes in either plus or minus $\boldsymbol{e}_{fied}^T\boldsymbol{X}_\mathcal{H}^t$.

## 6. Experimental evaluation.

We follow Farhadkhani et al. (2023) (on which the core of our code is based), and present results for classification tasks on MNIST and CIFAR-10 datasets, as well as plain averaging tasks. We refer to Appendix C for most of the experiments. Similarly to Farhadkhani et al. (2023), heterogeneity is simulated by sampling data from each class using a Dirichlet distribution of parameter $\alpha$. We test the attacks Spectral Heterogeneity (SpH), Dissensus, A Little Is Enough (ALIE) (Baruch et al., 2019), and Fall of Empire (FOE) (Xie et al., 2020). The main differences with Farhadkhani et al. (2023) are the following

(i) We consider sparse communication networks.
(ii) We implement GTS–RG instead of NNA, and ClippedGossip (aka CS$_{He}$–RG) is implemented the adaptive rule of clipping of (He et al., 2023) instead of a fixed threshold. We additionally implement IOS (Wu et al., 2023).
(iii) We add Dissensus and Spectral Heterogeneity attacks.
(iv) We modify the generic design of attacks to adapt it really to the decentralized setting.

See Appendix B for a detailed experimental setup and our implementation available at https://github.com/renaudgaucher/Byzantine-Robust-Gossip.

In Figure 1, it appears that the SpH attack is more efficient in disrupting ClippedGossip, GTS–RG and IOS than Dissensus and ALIE, and that CS$_+$–RG is highly resilient in the setup considered. In this setting with a rather simple learning task, the connectivity of the graphs appears as the major limiting factor to the robustness of distributed algorithms, hence why Spectral Heterogeneity is very efficient.

# 7. Conclusion

This paper revisits robust averaging over sparse communication graphs. We introduce a general framework for robust decentralized averaging, which allows us to derive tight convergence guarantees for many robust summation rules. In particular, we introduce one that nearly matches an upper bound on the breakdown point, *i.e.*, the maximum number of Byzantine nodes an algorithm can tolerate. Our experiments confirm that our theory correctly sorts the breakdown points of the existing methods, and that some (such as NNA) fail before the optimal breakdown point. We introduce a new *Spectral Heterogeneity* attack that exploits the graph topology for sparse graphs to obtain this result. An interesting future direction is the characterization of robustness when the constraint on the number of neighbors cannot be met globally, but convergence can be obtained within local neighborhoods. Conversely, this opens up questions on which nodes an attacker should corrupt to maximize their influence for a specific graph, in light of our results.

# Impact Statement

This paper presents work whose goal is to advance the field of Machine Learning. There are many potential societal consequences of our work, none which we feel must be specifically highlighted here.

# Acknowledgments

The work of Aymeric Dieuleveut and Renaud Gaucher was supported by French State aid managed by the Agence Nationale de la Recherche (ANR) under France 2030 program with the reference ANR-23-PEIA-005 (REDEEM project). The work of Aymeric Dieuleveut was also supported by Hi!Paris - FLAG chair.

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

# A. Additional Discussion

## A.1. Asynchronous communications

In the core of this paper, we assumed that all communications were synchronous. However, our F–RG framework can be readily adapted to operate in less synchronous settings.

Suppose that communication still occurs in rounds, but messages take a variable amount of time to be delivered. If each honest node waits to receive messages from all its neighbors before performing an aggregation step, then Byzantine nodes can prevent the honest nodes from updating simply by withholding their messages. To be robust against such behavior, the nodes should not wait for all messages before updating their parameters.

The F–RG framework adapts naturally to this setting. Specifically, consider a $(\rho, b)$-robust summand $F$, and assume that each honest node $i$ performs the F–RG update as soon as it has received all messages except those corresponding to a total weight of at most $b$. Then, this asynchronous version of F–RG remains $(1 - \eta(\mu_{\min} - 4\rho b))$-robust, as the following proposition demonstrates.

**Proposition A.1.** *Let $F : (\mathbb{R}_+ \times \mathbb{R}^d)^n \to \mathbb{R}^d$ be a $(b, \rho)$–robust summation. Let $F_{asyn.}$ denote the rule that applies $F$ to all inputs $(w_i, \boldsymbol{x}_i)_{i \in [n]}$ excluding an arbitrary subset $S_{delayed} \subset [n]$ of smaller than $b$, i.e. $\sum_{i \in S_{delayed}} w_i \leq b$. Then $F_{asyn.}$ is a $(b, 2\rho)$–robust summation rule.*

*Proof.* Using the triangle inequality, $(a + b)^2 \leq 2(a^2 + b^2)$ and Jensen inequality and the definition of robust summation yields

$$\left\| F_{asyn.}\big((\omega_i, \boldsymbol{z}_i)_{i \in [n]}\big) - \sum_{i \in S} \omega_i \boldsymbol{z}_i \right\|^2 = \left\| F\big((\omega_i, \boldsymbol{z}_i)_{i \in [n] \setminus S_{delayed}}\big) - \sum_{i \in [n] \setminus S_{delayed}} \omega_i \boldsymbol{z}_i - \sum_{i \in S \cap S_{delayed}} \omega_i \boldsymbol{z}_i \right\|^2$$

$$\leq \left( \left\| F\big((\omega_i, \boldsymbol{z}_i)_{i \in [n] \setminus S_{delayed}}\big) - \sum_{i \in [n] \setminus S_{delayed}} \omega_i \boldsymbol{z}_i \right\| + \left\| \sum_{i \in S \cap S_{delayed}} \omega_i \boldsymbol{z}_i \right\| \right)^2$$

$$\leq 2 \left( \left\| F\big((\omega_i, \boldsymbol{z}_i)_{i \in [n] \setminus S_{delayed}}\big) - \sum_{i \in [n] \setminus S_{delayed}} \omega_i \boldsymbol{z}_i \right\|^2 + \left\| b \sum_{i \in S \cap S_{delayed}} \frac{\omega_i}{b} \boldsymbol{z}_i \right\|^2 \right)$$

$$\leq 2 \left( \rho b \sum_{i \in S \setminus S_{delayed}} \omega_i \|\boldsymbol{z}_i\|^2 + b \sum_{i \in S \cap S_{delayed}} \omega_i \|\boldsymbol{z}_i\|^2 \right).$$

Now, since necessarily $\rho \geq 1$, it finally yields

$$\left\| F_{asyn.}\big((\omega_i, \boldsymbol{z}_i)_{i \in [n]}\big) - \sum_{i \in S} \omega_i \boldsymbol{z}_i \right\|^2 \leq 2\rho b \sum_{i \in S} \omega_i \|\boldsymbol{z}_i\|^2.$$

$\square$

*Remark* A.2. In the above result, we used the fact that there exists no $(b, \rho)$-robust summand with $\rho < 1$, as shown in Theorem 3.5.

*Remark* A.3. The latter proof can be refined using $(a + b)^2 \leq (1 + \epsilon)a^2 + (1 + \epsilon^{-1})b^2$ with an optimal choice of $\epsilon$, showing that $F_{asyn.}$ is at least $(b, \rho + \sqrt{\rho})$-robust.

## A.2. Choosing the graph's weights

Our analysis requires that each edge of the graph be associated with a nonnegative weight and that the communication step size satisfies $\eta \leq 1/\mu_{\max}(\mathcal{G}_{\mathcal{H}})$. Although unitary weights $w_{ij} = 1$ are convenient for identifying $b$ as an upper bound on the *number of Byzantine neighbors*, they require adjusting the step size $\eta$ using global information from the graph, such as $\mu_{\max}(\mathcal{G})^{-1}$. A practical way to circumvent this is to use bistochastic weights.

- **Generic Bistochastic Matrix.** Any bistochastic matrix $B \in [0,1]^{m \times m}$ can be used to define the weights of the graph using $w_{ij} = B_{ij}$. In this case, the Laplacian matrix is defined as $W = I - B$, and its largest eigenvalue is upper-bounded by 2. The communication step size can thus be chosen as $\eta = 1/2$. Note that, in such a case, the condition $\mu_2(\mathcal{G}_\mathcal{H}) \geq 2\rho b$ is implied by $\tilde{\gamma} \geq 2\rho b$, where $\tilde{\gamma}$ is generally named the *spectral gap* of the bistochastic matrix $B$, and is defined as $\tilde{\gamma} = 1 - \max_{\mu \in sp(B), \mu \neq 1} |\mu|$, where $sp(B)$ denotes the eigenvalues of the matrix $B$.

- **Metropolis-Hasting Weights ([Hastings, 1970](#)).** The Metropolis-Hasting algorithm constructs a bistochastic matrix by making each node $i$ declare to their neighbors their degree $d_i$. Then, any pair of neighbors $(i,j) \in \mathbb{E}$ defines the weight on their edge as $w_{ij} = 1/\max(d_i, d_j) + 1$. As pointed out in [He et al. (2023)](#) this algorithm is robust to corrupted nodes, since for $i \in \mathcal{H}$ and $j \in B$ the influence of $j$ on $i$ is bounded by $w_{ij} \leq \frac{1}{d_i+1}$. Interestingly, since the size of the communication step can be chosen as $\eta = 1/2$ without further knowledge of the global network properties, this choice of weights requires only local information to carry out the communication.

# B. Experiments

## B.1. Detailed Experimental Setup

### B.1.1. ATTACK DESIGN

Our experimental setting is built on top of the code provided by [Farhadkhani et al. (2023)](#), with the following differences:

1. Each honest node receives different messages from Byzantine nodes: for an honest node $i \in \mathcal{H}$, the Byzantine node $j \in n_\mathcal{B}(i)$ declares to node $i$ at time $t$ the vector $\boldsymbol{x}_j^t = \boldsymbol{x}_i^t + \zeta_i^t \boldsymbol{a}_i^t$. The reference point taken is the parameter of node $i$, instead of the average of all parameters $\overline{\boldsymbol{x}}_\mathcal{H}^t$, as performed in ([Farhadkhani et al., 2022](#)). Indeed, $\overline{\boldsymbol{x}}_\mathcal{H}^t$ can be very far from the vectors in the honest neighborhood of node $i$ since the network is not fully connected. Note that in opposition to ([Farhadkhani et al., 2022](#)), Byzantines declare *different* parameters to each of the honest nodes. Not only does it allow the use of attacks such as Dissensus and spectral heterogeneity (though the choice of $\boldsymbol{a}_i^t$), but it also allows to tune $\zeta_i^t$ differently for each node.

2. Each scaling parameter $\zeta_i^t$ is designed separately through a linear search, such as to maximize for each honest node $i$

$$\left\| F\big((w_{ij}, \boldsymbol{x}_i - \boldsymbol{x}_j)_{j \in n(i)}\big) - \sum_{j \in n_\mathcal{H}(j)} w_{ij}(\boldsymbol{x}_i - \boldsymbol{x}_j) \right\|^2.$$

3. The vector $\boldsymbol{a}_i^t$ is defined differently depending on the attack implemented: *Dissensus*, *Spectral Heterogeneity* (SpH), *Fall of Empire* (FOE) from [Xie et al. (2020)](#) or *A little is enough* (ALIE) from [Baruch et al. (2019)](#).

    - **Dissensus**. The Byzantines $j \in n_\mathcal{H}(i)$ take as attack vector $\boldsymbol{a}_i^t = [\boldsymbol{W}_\mathcal{H}^t \boldsymbol{X}_\mathcal{H}^t]_i = \sum_{j \in n_\mathcal{H}(i)} w_{ij}(\boldsymbol{x}_i^t - \boldsymbol{x}_j^t)$.
    - **Spectral Heterogeneity**. The Byzantines $j \in n_\mathcal{H}(i)$ take as attack vector $\boldsymbol{a}_i^t = [\boldsymbol{e}_{fied} \boldsymbol{e}_{fied}^T \boldsymbol{X}_\mathcal{H}^t]_i$, where $\boldsymbol{e}_{fied}$ denotes an eigenvector of $\boldsymbol{W}_\mathcal{H}$ associated with $\mu_2(\boldsymbol{W}_\mathcal{H})$.
    - **ALIE**. The Byzantine nodes compute the mean of the honest parameters $\overline{\boldsymbol{x}}_\mathcal{H}^t$ and the coordinate-wise standard deviation $\boldsymbol{\sigma}^t$. Then they use the attack vector $\boldsymbol{a}_i^t = \boldsymbol{\sigma}^t$.
    - **FOE**. The Byzantine nodes uses $\boldsymbol{a}_i^t = -\overline{\boldsymbol{x}}_\mathcal{H}^t$.

*Remark* B.1. In the case of trimming base rules, a badly designed attack leads Byzantine messages to be removed during aggregations, which induces the resulting algorithm to behave as a plain non-corrupted D-SGD algorithm. Thus, proposing non-over-confident experimental proofs of trimming-based aggregation requires a fine design of the attacks.

### B.1.2. ALGORITHMS TESTED

**Networks**. Two topologies of the honest subgraph are investigated: 1) A "Two Worlds" graph, i.e. $\mathcal{G}_\mathcal{H} := [\mathcal{G}_{m=13, k=8, c=1}]_\mathcal{H}$. 2) Randomly sampled Erdos-Renyi graphs, with a varying probability of edge presence $p$. All graphs are equipped with unitary weights on the edges, for simplicity.

**Communications.** We compare

- GTS-RG;
- CS$_+$-RG;
- The version of ClippedGossip proposed in He et al. (2023) with an adaptive clipping rule which is not supported by any theory. Precisely, ClippedGossip is equivalent to CS$_{\text{He}}$-RG, with CS$_{\text{He}} := \text{CS}(\cdot; \tau_{\text{He}})$, where, for $\|z_1\| \le \ldots \le \|z_n\|$,
  $$\tau_{\text{He}}\big((\omega_i, z_i)_{i \in [n]}\big) = \sqrt{\frac{1}{b} \sum_{i \le |n_{\mathcal{H}}(i)|} \omega_i \|z_i\|^2},$$
- Iterative Outlier scissors (IOS), from (Wu et al., 2023).

F-RG based methods uses $\eta = \mu_{\max}(\mathcal{G}_{\mathcal{H}})^{-1}$.

**Optimization.** All learning experiments implement Algorithm 1, and only the communication part changes among experiments. It follows that, even though IOS is not explicitly combined with momentum in (Wu et al., 2023), we still implement it with momentum. We do this to have a fairer comparison between communication routines since (Farhadkhani et al., 2023) showed that momentum is key for robustness.

### B.1.3. DATASET PRE-PROCESSING

MNIST images receive an input image normalization of mean $0.1307$ and standard deviation $0.3081$. The images of CIFAR-10 are horizontally flipped, and a per-channel normalization is applied with means $(0.4914, 0.4822, 0.4465)$, and standard deviation $(0.2023, 0.1994, 0.2010)$.

### B.1.4. DATA HETEROGENEITY

We simulate data heterogeneity in the correct nodes' datasets following the method of (Farhadkhani et al., 2023) by making nodes sample from each class of the considered dataset (MNIST or CIFAR-19) using a Dirichlet distribution of parameter $\alpha > 0$: the smallest $\alpha$, the more probable it is to sample from one class only.

### B.1.5. MODEL ARCHITECTURE AND HYPER PARAMETERS

To present the detailed architecture of the models used, we adopt the following compact notation:

L(#outputs) represents a **fully-connected linear layer**, C(output channels) represents a **2D-convolutional layer** of kernel size 3 and padding 1, R stands for **ReLU activation**, B stands for **batch-normalization**, and D represents **dropout** with probability 0.25, S stands for **log-softmax** and NLL for **negative log-likelihood loss**.

The architecture of the model used and the experimental setup are proposed in Table 1.

*Table 1.* Detailed experimental setting

| Dataset | MNIST | | CIFAR-10 |
|---|---|---|---|
| Model type | CNN | | CNN |
| Model architecture | C(16)-R-M-L(10)-S | | C(32)-B-R-M-C(64)-B-R-M-C(128)-B-R-D-L(128)-R-D-L(10)-S |
| Loss | NLL | | NLL |
| Batch size | 64 | | 64 |
| Learning rate | $\eta_{op} = 0.1$ | | $\eta_{op} = 0.5$ |
| Momentum | $\beta = 0.9$ | | $\beta = 0.99$ |
| Number of Iterations | $T = 300$ | | $T = 5000$ |
| Number of honest nodes | $\|\mathcal{H}\| = 26$ | $\|\mathcal{H}\| = 20$ | $\|\mathcal{H}\| = 16$ |
| Graph | Two Worlds | Erdös Renyi | Two Worlds |
| Graph parameter | $k = 8$ | $p \in [0.26, 1]$ | $k = 6$ |
| Data Heterogeneity | $\alpha = 1$ | $\alpha = 1$ | $\alpha = 5$ |
| Byzantine weight | $b \in \{1, \ldots, 11\}$ | $b = 3$ | $b = 3$ |
| Number of seeds | 1 | 1 | 1 |

# C. Experiments

In Appendix C.1 we provide experiments with two world graphs taken as a communication network, both on learning tasks with MNIST and CIFAR-10 datasets, and on an averaging problem. Both on the MNIST and averaging task, we a varying amount of Byzantine weight to investigate the empirical robustness of each algorithm.

In Appendix C.2 we provide experiments on Erdos Renyi networks on MNIST and an averaging task. We study here the influence of the connectivity of the network on the robustness by varying the probability of each pair of honest nodes being connected.

## C.1. Experiments with Two World graphs

For both the Averaging experiments and the MNIST experiments, we fixed the subgraph of honest nodes to be $\mathcal{G}_{\mathcal{H}} :=$ $[\mathcal{G}_{m=13,k=8,c=1}]_{\mathcal{H}}$, and the weight of Byzantine $b$ varies. Note that Theorem 3.5 predict that, on this graph, no algorithm can be $r$-robust when $b > 8$.

**Averaging experiments (Figure 2).** Recall that $\mathcal{G}_{\mathcal{H}} := [\mathcal{G}_{m=13,k=8,c=1}]_{\mathcal{H}}$, is built on two fully connected cliques of 13 honest nodes, which are then connected. Here we initialize the parameters of honest nodes with a $\mathcal{N}(\boldsymbol{u}, I_d/d)$ distribution $(d = 5)$, where $\boldsymbol{u}$ is equal to $+(5, 0, \ldots, 0)^T$ for one of the two clique, and equal to $-(5, 0, \ldots, 0)^T$ for the other one. For each setting $(b, \text{Communication Algorithm, Attack})$, we perform experiments with 6 random seeds. The error plotted correspond to the mean square error $\sum_{i \in \mathcal{H}} \|\boldsymbol{x}_i^t - \overline{\boldsymbol{x}}_{\mathcal{H}}^0\|^2 / \sum_{i \in \mathcal{H}} \|\boldsymbol{x}_i^0 - \overline{\boldsymbol{x}}_{\mathcal{H}}^0\|^2$ achieved after 100 communication steps. The line corresponds to the average value among seeds, while the confidence interval corresponds to the maximum and minimal values encountered.

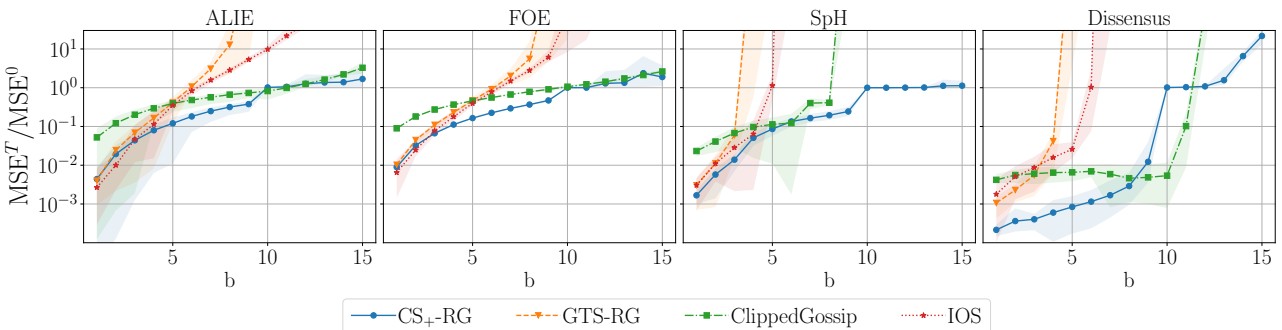

*Figure 2.* Relative MSE on an averaging task after 100 communication steps on a Two-Wold graph, with a varying weight of Byzantines $b$. Here $\mu_2(\mathcal{G}_{\mathcal{H}}) = 16$.

**MNIST experiments (Figure 3).** Experiments are conducted with a heterogeneity parameter $\alpha = 1$. The error and accuracy displayed are after 300 iterations. Further experimental details are in Table 1.

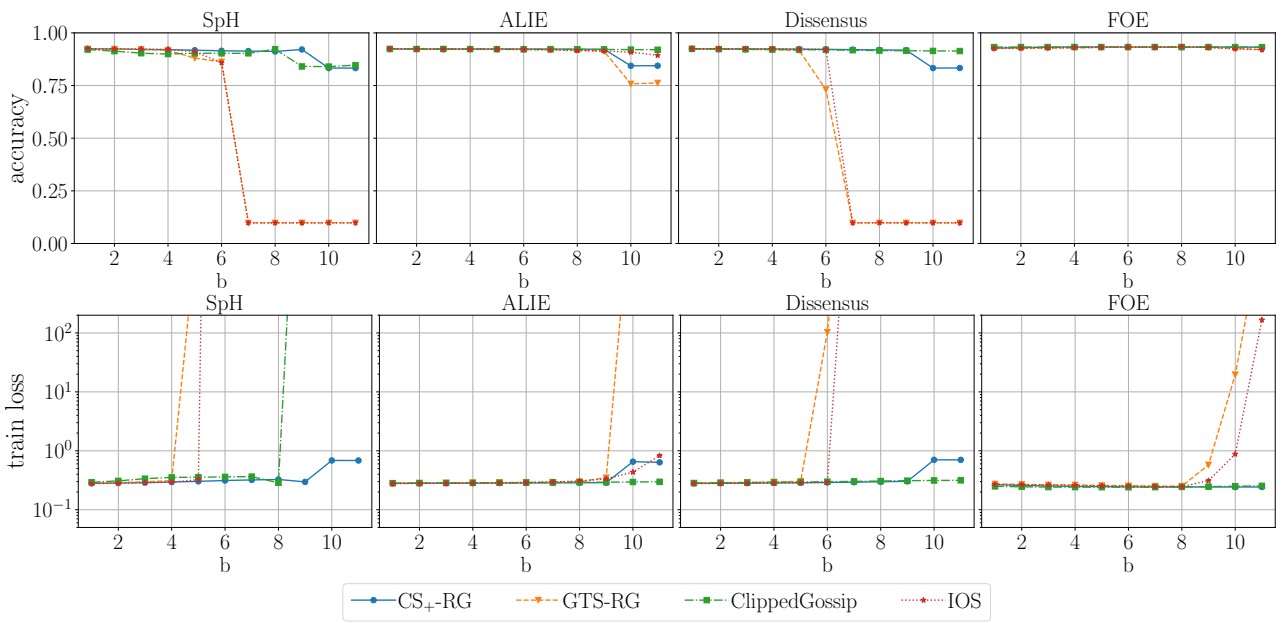

*Figure 3.* Accuracy and training loss on MNIST 300 optimization and communication steps, a Two Wold graphs, with a varying weight of Byzantines neighbors. Here $\mu_2(\mathcal{G}_{\mathcal{H}}) = 16$, and $\alpha = 1$.

**CIFAR-10 experiments (Figures 4 and 5).** Experiments are conducted on CIFAR-10 Dataset with an heterogeneity parameters $\alpha = 5$, on a Two Wold graph $\mathcal{G}_{\mathcal{H}} := [\mathcal{G}_{m=8,k=6,c=1}]_{\mathcal{H}}$ with $b = 1$. Further experimental details are in Table 1. We provide both test accuracy and train loss since the dynamic of the train loss does not always impact the test accuracy, specifically in the case of Spectral Heterogeneity and Dissensus attacks.

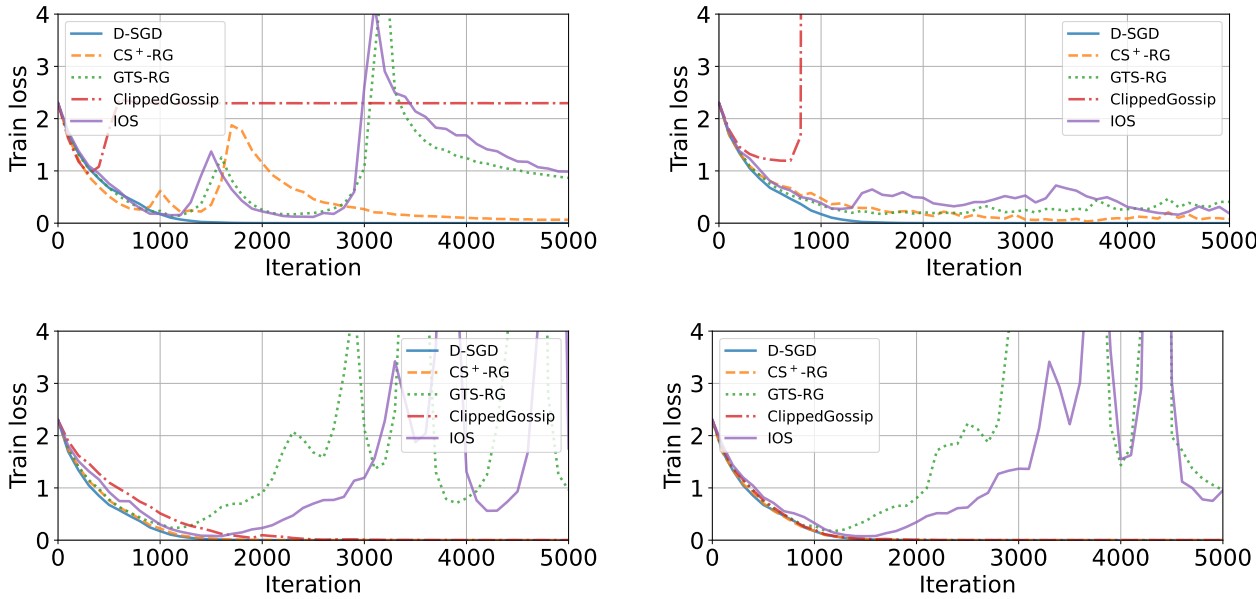

*Figure 4.* Train loss on CIFAR-10 ($\alpha = 5$) on a Two Wold graph with $\mu_2(\mathcal{G}_{\mathcal{H}}) = 12$ and $b = 1$. Attacks tested are FOE (upper left), ALIE (upper right), SpH (lower left), and Dissensus (lower right).

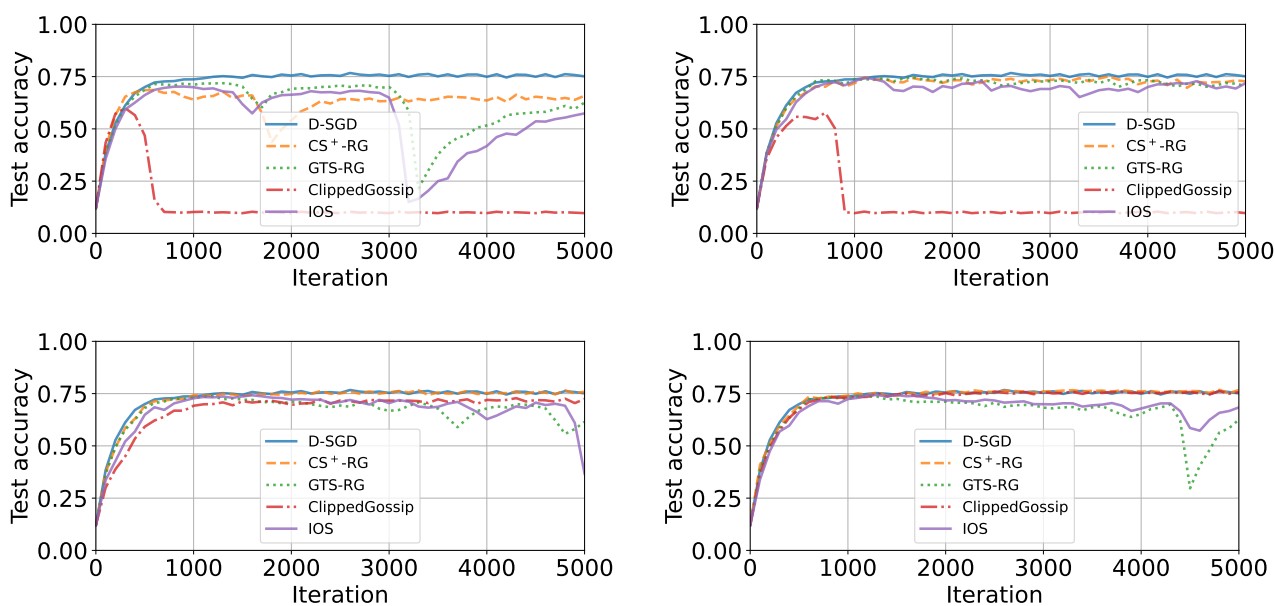

*Figure 5.* Accuracy on CIFAR-10 ($\alpha = 5$) on a Two Wold graph with $\mu_2(\mathcal{G}_\mathcal{H}) = 12$ and $b = 1$. Attacks tested are FOE (upper left), ALIE (upper right), SpH (lower left), and Dissensus (lower right).

### C.2. Experiments with Erdos-Renyi graphs

Experiments are conducted by using, as a subgraph of honest nodes, a random Erdos Renyi graph with 20 honest nodes. Each honest node is always adjacent to 4 Byzantine nodes. On each seed, we test 12 different values of $p \in [0.25, 1]$, where p denotes the probability of an edge to exist. We plot the links between the algebraic connectivity of the graph (denoted $\mu_2$, the second smallest eigenvalue of the unitary weighted Laplacian) and the losses.

**Averaging task (Figure 6).** Nodes' parameters are initialized using a $N(0, I_5)$ distribution. Nodes perform 100 (robust) gossip communication iterations, and the gain in terms of mean square error is plotted. Experiments are conducted on 6 different seeds, and curves are smoothed using a moving average of size 4.

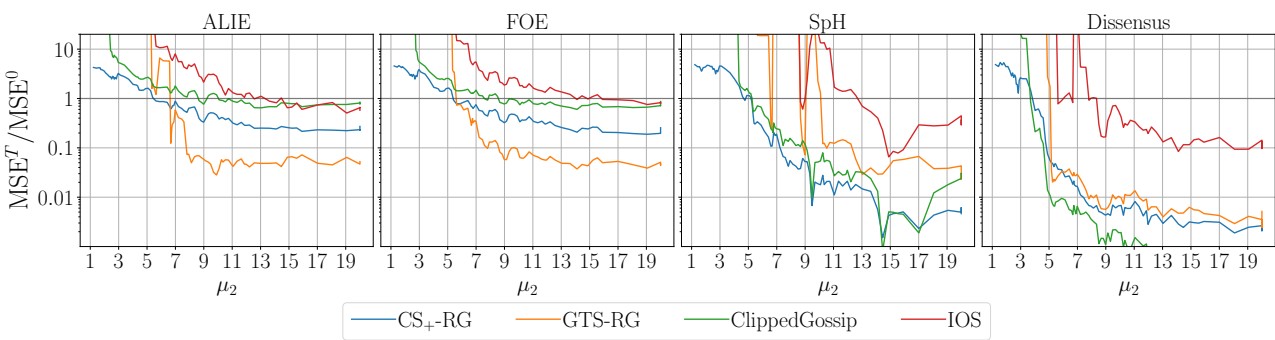

*Figure 6.* Averaging task. Relative MSE after 100 communication steps on randomly sampled Erdos-Renyi graphs with a fixed $b = 4$.

**MNIST (Figure 7).** The heterogeneity among nodes is set to $\alpha = 1$. All experiments run on the same seed.

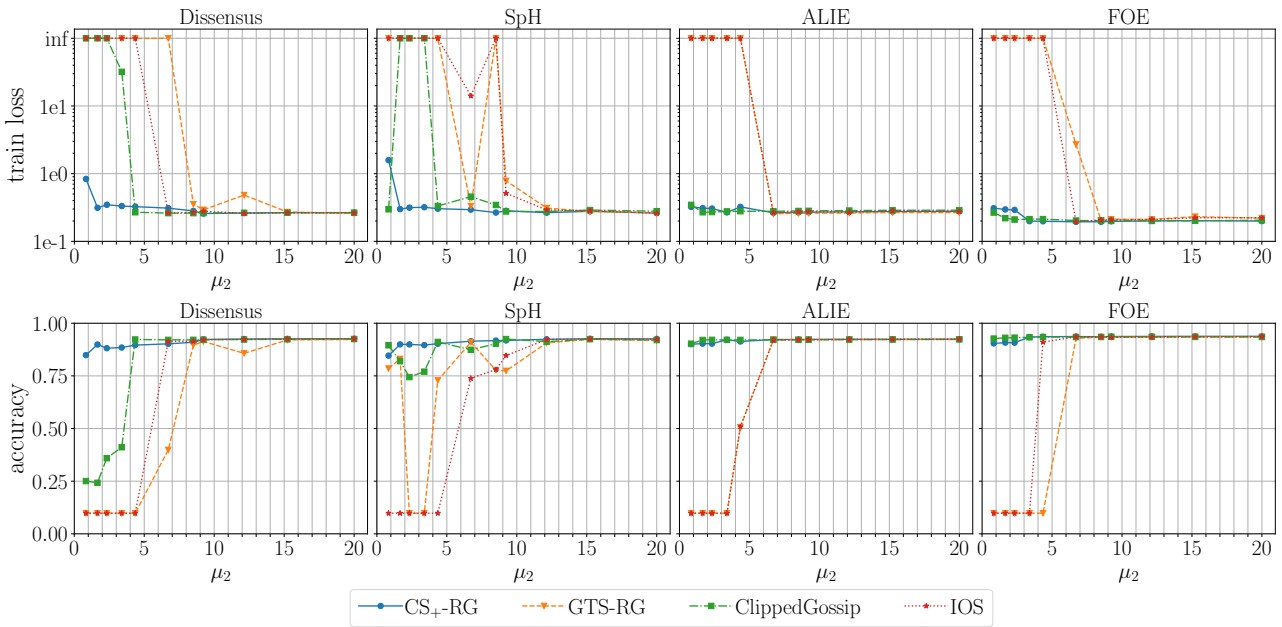

*Figure 7.* Accuracy and training loss on MNIST($\alpha = 1$) after 200 optimization and communication steps on randomly sampled Erdos-Renyi graphs with a fixed $b = 4$.

# D. Analysis of RG

## D.1. Proof of Theorem 3.3

We now prove Theorem 3.3, and then use it to derive convergence for the Byzantine-robust decentralized stochastic gradient descent framework. Recall that nodes follow the update scheme below.

$$\begin{cases} \boldsymbol{x}_i^{t+1} = \boldsymbol{x}_i^t - \eta F\left((w_{ij}; \boldsymbol{x}_j^t - \boldsymbol{x}_i^t)_{j \in n(i)}\right) & \text{if } i \in \mathcal{H} \\ \boldsymbol{x}_i^{t+1} = * & \text{if } i \in \mathcal{B}, \end{cases} \tag{5}$$

We recall the following notations

Before proving Theorem 3.3, we recall the following notations:

- $w_{ij} = -\boldsymbol{W}_{ij} \geq 0$ denote the weight associated with the edge $i \sim j$ on the graph.

- The matrix of honest parameters $\boldsymbol{X}_{\mathcal{H}}^t := \begin{pmatrix} (\boldsymbol{x}_1^t)^T \\ \vdots \\ (\boldsymbol{x}_{|\mathcal{H}|}^t)^T \end{pmatrix} \in \mathbb{R}^{|\mathcal{H}| \times d}$.

- The error due to robust aggregation and Byzantine corruption:

$$\forall i \in \mathcal{H}, \quad [\boldsymbol{E}^t]_i := \sum_{j \in n_{\mathcal{H}}(i)} w_{ij}(\boldsymbol{x}_i^t - \boldsymbol{x}_j^t) - F\left((w_{ij}; \boldsymbol{x}_j^t - \boldsymbol{x}_i^t)_{j \in n(i)}\right)$$

**Lemma D.1.** *Equation* (5) *writes*

$$\boldsymbol{X}_{\mathcal{H}}^{t+1} = (\boldsymbol{I}_{\mathcal{H}} - \eta \boldsymbol{W}_{\mathcal{H}})\boldsymbol{X}_{\mathcal{H}}^t + \eta \boldsymbol{E}^t.$$

*Proof.* Let $i \in \mathcal{H}$. We decompose the update due to the gossip scheme and consider the error term coming from both robust aggregation and the influence of Byzantine nodes.

$$\boldsymbol{x}_i^{t+1} = \boldsymbol{x}_i^t - \eta F\left((w_{ij}; \boldsymbol{x}_j^t - \boldsymbol{x}_i^t)_{j \in n(i)}\right)$$

$$= \boldsymbol{x}_i^t - \eta \sum_{j \in n_{\mathcal{H}}(i)} w_{ij}(\boldsymbol{x}_i^t - \boldsymbol{x}_j^t) + \eta \left(\sum_{j \in n_{\mathcal{H}}(i)} w_{ij}(\boldsymbol{x}_i^t - \boldsymbol{x}_j^t) - F\left((w_{ij}; \boldsymbol{x}_j^t - \boldsymbol{x}_i^t)_{j \in n(i)}\right)\right).$$

Finally, the proof is concluded by remarking that $[W_{\mathcal{H}} X_{\mathcal{H}}^t]_i = \sum_{j \in n_{\mathcal{H}}(i)} w_{ij}(\boldsymbol{x}_i^t - \boldsymbol{x}_j^t)$. $\qquad \square$

We begin by controlling the norm of the error term $\|\boldsymbol{E}^t\|_2^2$ in the case of CS$_+$–RG.

**Lemma D.2** (Control of the error). *Assume $F$ is a $(b, \rho)$ robust summation. Then the error is controlled by the heterogeneity as measured by the Laplacian matrix:*

$$\|\boldsymbol{E}^t\|_2^2 \le 2\rho b \|\boldsymbol{X}_{\mathcal{H}}^t\|_{\boldsymbol{W}_{\mathcal{H}}}^2 = \rho b \sum_{i \in \mathcal{H}, j \in n_{\mathcal{H}}(i)} w_{ij} \|\boldsymbol{x}_i^t - \boldsymbol{x}_j^t\|^2.$$

*Proof.* We recall that in this case,

$$\forall i \in \mathcal{H}, \quad [\boldsymbol{E}^t]_i := \sum_{j \in n_{\mathcal{H}}(i)} w_{ij}(\boldsymbol{x}_i^t - \boldsymbol{x}_j^t) - F\left((w_{ij}; \boldsymbol{x}_j^t - \boldsymbol{x}_i^t)_{j \in n(i)}\right).$$

By assumption, for all honest node $i \in \mathcal{H}$, the weight of Byzantine in his neighborhood is smaller than $b$, i.e $\sum_{j \in n_{\mathcal{B}}(i)} w_{ij} \le b$. Thus, applying the $(b, \rho)$ robustness of F yields

$$\|\boldsymbol{E}^t\|^2 = \sum_{i \in \mathcal{H}} \left\| \sum_{j \in n_{\mathcal{H}}(i)} w_{ij}(\boldsymbol{x}_i^t - \boldsymbol{x}_j^t) - F\left((w_{ij}; \boldsymbol{x}_j^t - \boldsymbol{x}_i^t)_{j \in n(i)}\right) \right\|_2^2$$

$$\le \sum_{i \in \mathcal{H}} \rho b \sum_{j \in n_{\mathcal{H}}(i)} w_{ij} \|\boldsymbol{x}_i^t - \boldsymbol{x}_j^t\|^2$$

$$= 2\rho b \|\boldsymbol{X}_{\mathcal{H}}^t\|_{\boldsymbol{W}_{\mathcal{H}}}^2,$$

Where the last equality follows by noting that $2\|\boldsymbol{X}_{\mathcal{H}}^t\|_{\boldsymbol{W}_{\mathcal{H}}}^2 = \sum_{i \in \mathcal{H}, j \in n_{\mathcal{H}}(i)} w_{ij} \|\boldsymbol{x}_i^t - \boldsymbol{x}_j^t\|^2$. Indeed, considering that $\mathcal{G}_{\mathcal{H}}$ is an undirected graph, $i \in n_{\mathcal{H}}(j) \iff j \in n_{\mathcal{H}}(i)$ and we have:

$$\|\boldsymbol{X}_{\mathcal{H}}^t\|_{\boldsymbol{W}_{\mathcal{H}}}^2 = \langle \boldsymbol{X}_{\mathcal{H}}, \boldsymbol{W}_{\mathcal{H}} \boldsymbol{X}_{\mathcal{H}} \rangle$$

$$= \sum_{i \in \mathcal{H}} \left\langle \boldsymbol{x}_i^t, \sum_{j \in n_{\mathcal{H}}(i)} w_{ij}(\boldsymbol{x}_i^t - \boldsymbol{x}_j^t) \right\rangle$$

$$= \sum_{i \in \mathcal{H}} \sum_{j \in n_{\mathcal{H}}(i)} w_{ij} \left\langle \boldsymbol{x}_i^t, \boldsymbol{x}_i^t - \boldsymbol{x}_j^t \right\rangle$$

$$= \frac{1}{2} \sum_{i \in \mathcal{H}} \sum_{j \in n_{\mathcal{H}}(i)} w_{ij} \left\langle \boldsymbol{x}_i^t - \boldsymbol{x}_j^t, \boldsymbol{x}_i^t - \boldsymbol{x}_j^t \right\rangle$$

$$\|\boldsymbol{X}_{\mathcal{H}}^t\|_{\boldsymbol{W}_{\mathcal{H}}}^2 = \frac{1}{2} \sum_{i \in \mathcal{H}, j \in n_{\mathcal{H}}(i)} w_{ij} \left\| \boldsymbol{x}_i^t - \boldsymbol{x}_j^t \right\|_2^2.$$

$\qquad \square$

Now that we control the error term, we can conclude the proof of Theorem 3.3 using standard optimization arguments. Before proving this theorem, we prove the following one, from which Corollary 3.4 is direct.

**Theorem D.3.** *Assume $F$ is a $(b, \rho)$ robust summation, and $\mathrm{RG}$ is the associated robust gossip algorithm. Let $b$ and $\mu_{\min}$ be such that $2\rho b \leq \mu_{\min}$, and let $\mathcal{G} \in \Gamma_{\mu_{\min}, b}$. Then, for any $\eta \leq \mu_{\max}(\mathcal{G}_{\mathcal{H}})^{-1}$, the output $\boldsymbol{y} = \mathrm{RG}(\boldsymbol{x})$ (obtained by one step of $\mathrm{RG}$ on $\mathcal{G}$ from $\boldsymbol{x}$) verifies:*

$$\frac{1}{|\mathcal{H}|} \sum_{i \in \mathcal{H}} \|\boldsymbol{x}_i^{t+1} - \overline{\boldsymbol{x}}_{\mathcal{H}}^{t+1}\|^2 \leq (1 - \eta(\mu_{\min} - 2\rho b)) \frac{1}{|\mathcal{H}|} \sum_{i \in \mathcal{H}} \|\boldsymbol{x}_i^t - \overline{\boldsymbol{x}}_{\mathcal{H}}^t\|^2 \tag{6}$$

$$\|\overline{\boldsymbol{x}}_{\mathcal{H}}^{t+1} - \overline{\boldsymbol{x}}_{\mathcal{H}}^t\|^2 \leq \eta \frac{2\rho b}{|\mathcal{H}|} \sum_{i \in \mathcal{H}} \|\boldsymbol{x}_i^t - \overline{\boldsymbol{x}}_{\mathcal{H}}^t\|^2. \tag{7}$$

*Proof.* **Part I: Equation (7).**

Equation (7) is a direct consequence of Lemma D.2. Indeed applying $\boldsymbol{P}_{\boldsymbol{1}_{\mathcal{H}}} := \frac{1}{|\mathcal{H}|} \boldsymbol{1}_{\mathcal{H}} \boldsymbol{1}_{\mathcal{H}}^T$ - the orthogonal projection on the kernel of $\boldsymbol{W}_{\mathcal{H}}$ - on Lemma D.1 results in

$$\boldsymbol{P}_{\boldsymbol{1}_{\mathcal{H}}} \boldsymbol{X}_{\mathcal{H}}^{t+1} = \boldsymbol{P}_{\boldsymbol{1}_{\mathcal{H}}} (\boldsymbol{I}_{\mathcal{H}} - \eta \boldsymbol{W}_{\mathcal{H}}) \boldsymbol{X}_{\mathcal{H}}^t + \eta \boldsymbol{P}_{\boldsymbol{1}_{\mathcal{H}}} \boldsymbol{E}^t = \boldsymbol{P}_{\boldsymbol{1}_{\mathcal{H}}} \boldsymbol{X}_{\mathcal{H}}^t + \eta \boldsymbol{P}_{\boldsymbol{1}_{\mathcal{H}}} \boldsymbol{E}^t.$$

Taking the norm yields

$$\|\boldsymbol{P}_{\boldsymbol{1}_{\mathcal{H}}} \boldsymbol{X}_{\mathcal{H}}^{t+1} - \boldsymbol{P}_{\boldsymbol{1}_{\mathcal{H}}} \boldsymbol{X}_{\mathcal{H}}^t\|^2 = \eta^2 \|\boldsymbol{P}_{\boldsymbol{1}_{\mathcal{H}}} \boldsymbol{E}^t\|^2 \leq \eta^2 \|\boldsymbol{E}^t\|^2. \tag{8}$$

We now apply Lemma D.2, and use that $\mu_{\max}(\mathcal{G}_{\mathcal{H}})$ is the largest eigenvalue of $\boldsymbol{W}_{\mathcal{H}}$. It gives

$$\|\boldsymbol{P}_{\boldsymbol{1}_{\mathcal{H}}} \boldsymbol{X}_{\mathcal{H}}^{t+1} - \boldsymbol{P}_{\boldsymbol{1}_{\mathcal{H}}} \boldsymbol{X}_{\mathcal{H}}^t\|^2 \leq \eta^2 2\rho b \|\boldsymbol{X}_{\mathcal{H}}^t\|_{\boldsymbol{W}_{\mathcal{H}}}^2$$
$$\leq \mu_{\max}(\mathcal{G}_{\mathcal{H}}) \eta^2 2\rho b \|(\boldsymbol{I}_{\mathcal{H}} - \boldsymbol{P}_{\boldsymbol{1}_{\mathcal{H}}}) \boldsymbol{X}_{\mathcal{H}}^t\|^2$$

Finally, Equation (7) derives from $[\boldsymbol{P}_{\boldsymbol{1}_{\mathcal{H}}} \boldsymbol{X}_{\mathcal{H}}^t]_{i \in \mathcal{H}} = [\sum_{j \in \mathcal{H}} \boldsymbol{x}_j^t]_{i \in \mathcal{H}} = [\overline{\boldsymbol{x}}_{\mathcal{H}}^t]_{i \in \mathcal{H}}$ and $\eta \mu_{\max}(\mathcal{G}_{\mathcal{H}}) \leq 1$.

**Part II: Equation (6).**

To prove Equation (6), we consider the objective function $\|(\boldsymbol{I}_{\mathcal{H}} - \boldsymbol{P}_{\boldsymbol{1}_{\mathcal{H}}}) \boldsymbol{X}^t\|^2$. We denote by $\boldsymbol{W}_{\mathcal{H}}^\dagger$ the Moore-Penrose pseudo inverse of $\boldsymbol{W}_{\mathcal{H}}$. We begin by applying Lemma D.1.

$$\|(\boldsymbol{I}_{\mathcal{H}} - \boldsymbol{P}_{\boldsymbol{1}_{\mathcal{H}}}) \boldsymbol{X}_{\mathcal{H}}^{t+1}\|^2 = \|\boldsymbol{X}_{\mathcal{H}}^t - \eta \boldsymbol{W}_{\mathcal{H}} \boldsymbol{X}_{\mathcal{H}}^t + \eta \boldsymbol{E}^t\|_{(\boldsymbol{I}_{\mathcal{H}} - \boldsymbol{P}_{\boldsymbol{1}_{\mathcal{H}}})}^2$$
$$= \|\boldsymbol{X}_{\mathcal{H}}^t\|_{(\boldsymbol{I}_{\mathcal{H}} - \boldsymbol{P}_{\boldsymbol{1}_{\mathcal{H}}})}^2 - 2\eta \left\langle \boldsymbol{X}_{\mathcal{H}}^t, \boldsymbol{W}_{\mathcal{H}} \boldsymbol{X}_{\mathcal{H}}^t - \boldsymbol{E}^t \right\rangle_{(\boldsymbol{I}_{\mathcal{H}} - \boldsymbol{P}_{\boldsymbol{1}_{\mathcal{H}}})} + \eta^2 \left\| \boldsymbol{W}_{\mathcal{H}} \boldsymbol{X}_{\mathcal{H}}^t - \boldsymbol{E}^t \right\|_{(\boldsymbol{I}_{\mathcal{H}} - \boldsymbol{P}_{\boldsymbol{1}_{\mathcal{H}}})}$$
$$= \|\boldsymbol{X}_{\mathcal{H}}^t\|_{(\boldsymbol{I}_{\mathcal{H}} - \boldsymbol{P}_{\boldsymbol{1}_{\mathcal{H}}})}^2 - 2\eta \left\langle \boldsymbol{X}_{\mathcal{H}}^t, \boldsymbol{X}_{\mathcal{H}}^t - \boldsymbol{W}_{\mathcal{H}}^\dagger \boldsymbol{E}^t \right\rangle_{\boldsymbol{W}_{\mathcal{H}}} + \eta^2 \left\| \boldsymbol{X}_{\mathcal{H}}^t - \boldsymbol{W}_{\mathcal{H}}^\dagger \boldsymbol{E}^t \right\|_{\boldsymbol{W}_{\mathcal{H}}^2}.$$

Applying $2\langle \boldsymbol{a}, \boldsymbol{b} \rangle = \|\boldsymbol{a}\|^2 + \|\boldsymbol{b}\|^2 - \|\boldsymbol{a} - \boldsymbol{b}\|^2$ leads to

$$\|\boldsymbol{X}_{\mathcal{H}}^{t+1}\|_{(\boldsymbol{I}_{\mathcal{H}} - \boldsymbol{P}_{\boldsymbol{1}_{\mathcal{H}}})}^2 - \|\boldsymbol{X}_{\mathcal{H}}^t\|_{(\boldsymbol{I}_{\mathcal{H}} - \boldsymbol{P}_{\boldsymbol{1}_{\mathcal{H}}})}^2 = -\eta \|\boldsymbol{X}_{\mathcal{H}}^t\|_{\boldsymbol{W}_{\mathcal{H}}}^2 - \eta \left\| \boldsymbol{X}_{\mathcal{H}}^t - \boldsymbol{W}_{\mathcal{H}}^\dagger \boldsymbol{E}^t \right\|_{\boldsymbol{W}_{\mathcal{H}}}^2 + \eta \left\| \boldsymbol{W}_{\mathcal{H}}^\dagger \boldsymbol{E}^t \right\|_{\boldsymbol{W}_{\mathcal{H}}}^2 \tag{9}$$
$$+ \eta^2 \left\| \boldsymbol{X}_{\mathcal{H}}^t - \boldsymbol{W}_{\mathcal{H}}^\dagger \boldsymbol{E}^t \right\|_{\boldsymbol{W}_{\mathcal{H}}^2}$$
$$= -\eta \|\boldsymbol{X}_{\mathcal{H}}^t\|_{\boldsymbol{W}_{\mathcal{H}}}^2 + \eta \|\boldsymbol{E}^t\|_{\boldsymbol{W}_{\mathcal{H}}^\dagger}^2 - \eta \left\| \boldsymbol{X}_{\mathcal{H}}^t - \boldsymbol{W}_{\mathcal{H}}^\dagger \boldsymbol{E}^t \right\|_{\boldsymbol{W}_{\mathcal{H}}}^2 + \eta^2 \left\| \boldsymbol{X}_{\mathcal{H}}^t - \boldsymbol{W}_{\mathcal{H}}^\dagger \boldsymbol{E}^t \right\|_{\boldsymbol{W}_{\mathcal{H}}^2}.$$

We now apply that $\mu_{\max}(\mathcal{G}_{\mathcal{H}})$ (resp. $\mu_2(\mathcal{G}_{\mathcal{H}})$) is the largest (resp. smallest) non-zero eigenvalue of $\boldsymbol{W}_{\mathcal{H}}$.

$$\|\boldsymbol{X}_{\mathcal{H}}^{t+1}\|_{(\boldsymbol{I}_{\mathcal{H}} - \boldsymbol{P}_{\boldsymbol{1}_{\mathcal{H}}})}^2 - \|\boldsymbol{X}_{\mathcal{H}}^t\|_{(\boldsymbol{I}_{\mathcal{H}} - \boldsymbol{P}_{\boldsymbol{1}_{\mathcal{H}}})}^2 \leq -\eta \left\| \boldsymbol{X}_{\mathcal{H}}^t \right\|_{\boldsymbol{W}_{\mathcal{H}}}^2 + \eta \frac{1}{\mu_2(\mathcal{G}_{\mathcal{H}})} \left\| \boldsymbol{E}^t \right\|^2 - \eta(1 - \mu_{\max}(\mathcal{G}_{\mathcal{H}})\eta) \left\| \boldsymbol{X}_{\mathcal{H}}^t - \boldsymbol{W}_{\mathcal{H}}^\dagger \boldsymbol{E}^t \right\|_{\boldsymbol{W}_{\mathcal{H}}}^2.$$

Eventually Lemma D.2 with the assumption $\eta \leq 1/\mu_{\max}(\mathcal{G}_{\mathcal{H}})$ yields the result

$$\|\boldsymbol{X}_{\mathcal{H}}^{t+1}\|_{(\boldsymbol{I}_{\mathcal{H}} - \boldsymbol{P}_{\boldsymbol{1}_{\mathcal{H}}})}^2 \leq \|\boldsymbol{X}_{\mathcal{H}}^t\|_{(\boldsymbol{I}_{\mathcal{H}} - \boldsymbol{P}_{\boldsymbol{1}_{\mathcal{H}}})}^2 - \eta \left( 1 - \frac{2\rho b}{\mu_2(\mathcal{G}_{\mathcal{H}})} \right) \|\boldsymbol{X}_{\mathcal{H}}^t\|_{\boldsymbol{W}_{\mathcal{H}}}^2$$

$$\|\boldsymbol{X}_{\mathcal{H}}^{t+1}\|_{(\boldsymbol{I}_{\mathcal{H}}-\boldsymbol{P}_{\mathbf{1}_{\mathcal{H}}})}^2 \le \left(1 - \eta\mu_2(\mathcal{G}_{\mathcal{H}})\left(1 - \frac{2\rho b}{\mu_2(\mathcal{G}_{\mathcal{H}})}\right)\right)\|\boldsymbol{X}_{\mathcal{H}}^t\|_{(\boldsymbol{I}_{\mathcal{H}}-\boldsymbol{P}_{\mathbf{1}_{\mathcal{H}}})}^2.$$

$\square$

To obtain Theorem D.3, we note that we can actually control the one-step variation of the MSE using $(1 - \eta(\mu_{\min} - 2\rho b))$ only, thus strengthening the first inequality. We rewrite the first part of Theorem 3.3 below for completeness.

**Corollary D.4.** *Assume $F$ is a $(b, \rho)$ robust summation, and RG is the associated robust gossip algorithm. Let $b$ and $\mu_{\min}$ be such that $2\rho b \le \mu_{\min}$, and let $\mathcal{G} \in \Gamma_{\mu_{\min}, b}$. Then, for any $\eta \le \mu_{\max}(\mathcal{G}_{\mathcal{H}})^{-1}$, the output $\boldsymbol{y} = \text{RG}(\boldsymbol{x})$ (obtained by one step of RG on $\mathcal{G}$ from $\boldsymbol{x}$) verifies:*

$$\frac{1}{|\mathcal{H}|}\sum_{i\in\mathcal{H}}\|\boldsymbol{x}_i^{t+1} - \overline{\boldsymbol{x}}_{\mathcal{H}}^t\|^2 \le (1 - \eta(\mu_{\min} - 2\rho b))\frac{1}{|\mathcal{H}|}\sum_{i\in\mathcal{H}}\|\boldsymbol{x}_i^t - \overline{\boldsymbol{x}}_{\mathcal{H}}^t\|^2$$

*Proof.* We consider Equation (8) and Equation (9), which write

$$\|\boldsymbol{P}_{\mathbf{1}_{\mathcal{H}}}\boldsymbol{X}_{\mathcal{H}}^{t+1} - \boldsymbol{P}_{\mathbf{1}_{\mathcal{H}}}\boldsymbol{X}_{\mathcal{H}}^t\|^2 = \eta^2\|\boldsymbol{P}_{\mathbf{1}_{\mathcal{H}}}\boldsymbol{E}^t\|^2.$$

$$\|\boldsymbol{X}_{\mathcal{H}}^{t+1}\|_{(\boldsymbol{I}_{\mathcal{H}}-\boldsymbol{P}_{\mathbf{1}_{\mathcal{H}}})}^2 - \|\boldsymbol{X}_{\mathcal{H}}^t\|_{(\boldsymbol{I}_{\mathcal{H}}-\boldsymbol{P}_{\mathbf{1}_{\mathcal{H}}})}^2 \le -\eta\left\|\boldsymbol{X}_{\mathcal{H}}^t\right\|_{\boldsymbol{W}_{\mathcal{H}}}^2 + \eta\left\|\boldsymbol{E}^t\right\|_{\boldsymbol{W}_{\mathcal{H}}^\dagger}^2.$$

It follows from the bias - variance decomposition of the MSE

$$\|\boldsymbol{X}_{\mathcal{H}}^{t+1} - \boldsymbol{P}_{\mathbf{1}_{\mathcal{H}}}\boldsymbol{X}_{\mathcal{H}}^t\|^2 = \|(\boldsymbol{I}_{\mathcal{H}} - \boldsymbol{P}_{\mathbf{1}_{\mathcal{H}}})\boldsymbol{X}_{\mathcal{H}}^{t+1}\|^2 + \|\boldsymbol{P}_{\mathbf{1}_{\mathcal{H}}}\boldsymbol{X}_{\mathcal{H}}^{t+1} - \boldsymbol{P}_{\mathbf{1}_{\mathcal{H}}}\boldsymbol{X}_{\mathcal{H}}^t\|^2$$

that

$$\|\boldsymbol{X}_{\mathcal{H}}^{t+1} - \boldsymbol{P}_{\mathbf{1}_{\mathcal{H}}}\boldsymbol{X}_{\mathcal{H}}^t\|^2 - \|\boldsymbol{X}_{\mathcal{H}}^t\|_{(\boldsymbol{I}_{\mathcal{H}}-\boldsymbol{P}_{\mathbf{1}_{\mathcal{H}}})}^2 \le -\eta\left\|\boldsymbol{X}_{\mathcal{H}}^t\right\|_{\boldsymbol{W}_{\mathcal{H}}}^2 + \eta\left\|\boldsymbol{E}^t\right\|_{\boldsymbol{W}_{\mathcal{H}}^\dagger}^2 + \eta^2\|\boldsymbol{P}_{\mathbf{1}_{\mathcal{H}}}\boldsymbol{E}^t\|^2$$

$$\le -\eta\left\|\boldsymbol{X}_{\mathcal{H}}^t\right\|_{\boldsymbol{W}_{\mathcal{H}}}^2 + \eta\frac{1}{\mu_2(\mathcal{G}_{\mathcal{H}})}\left\|\boldsymbol{E}^t\right\|_{(\boldsymbol{I}_{\mathcal{H}}-\boldsymbol{P}_{\mathbf{1}_{\mathcal{H}}})}^2 + \eta^2\|\boldsymbol{P}_{\mathbf{1}_{\mathcal{H}}}\boldsymbol{E}^t\|^2$$

As $\eta \le \frac{1}{\mu_{\max}(\mathcal{G}_{\mathcal{H}})} \le \frac{1}{\mu_2(\mathcal{G}_{\mathcal{H}})}$, we eventually get

$$\|\boldsymbol{X}_{\mathcal{H}}^{t+1} - \boldsymbol{P}_{\mathbf{1}_{\mathcal{H}}}\boldsymbol{X}_{\mathcal{H}}^t\|^2 \le \|\boldsymbol{X}_{\mathcal{H}}^t\|_{(\boldsymbol{I}_{\mathcal{H}}-\boldsymbol{P}_{\mathbf{1}_{\mathcal{H}}})}^2 - \eta\left\|\boldsymbol{X}_{\mathcal{H}}^t\right\|_{\boldsymbol{W}_{\mathcal{H}}}^2 + \eta\frac{1}{\mu_2(\mathcal{G}_{\mathcal{H}})}\left\|\boldsymbol{E}^t\right\|^2$$

$$\le \|\boldsymbol{X}_{\mathcal{H}}^t\|_{(\boldsymbol{I}_{\mathcal{H}}-\boldsymbol{P}_{\mathbf{1}_{\mathcal{H}}})}^2 - \eta\left(1 - \frac{2\rho b}{\mu_2(\mathcal{G}_{\mathcal{H}})}\right)\left\|\boldsymbol{X}_{\mathcal{H}}^t\right\|_{\boldsymbol{W}_{\mathcal{H}}}^2$$

$$\le \left(1 - \eta\mu_2(\mathcal{G}_{\mathcal{H}})\left(1 - \frac{2\rho b}{\mu_2(\mathcal{G}_{\mathcal{H}})}\right)\right)\|\boldsymbol{X}_{\mathcal{H}}^t\|_{(\boldsymbol{I}_{\mathcal{H}}-\boldsymbol{P}_{\mathbf{1}_{\mathcal{H}}})}^2.$$

Which concludes the proof.  $\square$

### D.2. Proof of Corollary 3.4

A direct consequence of the above results is Corollary 3.4, as we show below.

*Proof.* Using the $(\alpha, \lambda)$ reduction notations, we have:

$$\begin{cases} \alpha = 1 - \gamma(1 - \delta)) \\ \lambda = \gamma\delta. \end{cases}$$

We denote here the drift increment $d_{t+1} = \|\boldsymbol{P}_{\mathbf{1}_{\mathcal{H}}}\boldsymbol{X}_{\mathcal{H}}^{t+1} - \boldsymbol{P}_{\mathbf{1}_{\mathcal{H}}}\boldsymbol{X}_{\mathcal{H}}^t\|$ and the variance at time $t$ as $\sigma_t^2 = \|\boldsymbol{X}_{\mathcal{H}}^{t+1}\|_{(\boldsymbol{I}_{\mathcal{H}}-\boldsymbol{P}_{\mathbf{1}_{\mathcal{H}}})}^2$.

Corollary D.4 ensures that

$$\sigma_{t+1}^2 + d_t^2 \le \alpha\sigma_t^2.$$

Hence, we have $\sigma_{t+1}^2 + d_{t+1}^2 \leq \alpha\sigma_t^2$, and so $\sigma_{t+1}^2 \leq \alpha\sigma_t^2$, which implies that $\sigma_t \leq \alpha^{t/2}\sigma_0$. This proves the first part of the result. Using this, we write that Theorem D.3 ensures that

$$d_{t+1} \leq \sqrt{\lambda}\sigma_t \leq \sqrt{\lambda}\beta^t\sigma 0,$$

leading to:

$$\sum_{t=1}^{T} d_t \leq \sqrt{\lambda}\sum_{t=0}^{T-1}\sigma_t \leq \sqrt{\lambda}\sum_{t=0}^{T-1}\alpha^{t/2}\sigma_0 \leq \frac{\sqrt{\lambda}(1-\alpha^{T/2})}{1-\alpha^{1/2}}\sigma 0,$$

which proves the second part. The last inequality is obtained by writing.

$$\|\boldsymbol{P_{1_\mathcal{H}}}\boldsymbol{X}_\mathcal{H}^T - \boldsymbol{P_{1_\mathcal{H}}}\boldsymbol{X}_\mathcal{H}^0\| \leq \sum_{t=1}^{T} d_t \leq \frac{\sqrt{\lambda}}{1-\sqrt{\alpha}}\sigma_0.$$

Then, we use that $0 \leq \frac{1}{1-\sqrt{1-x}} \leq \frac{2}{x}$ for $x \geq 0$, with $x = \gamma(1-\delta)$.

$\square$

# E. Proof of Theorem 3.5 - Upper bound on the breakdown point

We recall Theorem 3.5:

**Theorem E.1.** *Let $\mu_{\min} \geq 0$, $b \geq 0$ be such that $\mu_{\min} \leq 2b$. Then for any $h \geq 0$ and any algorithm Alg, there exists a graph $\mathcal{G} \in \Gamma_{\mu_{\min},b}$ in which all honest nodes have a weight of honest neighbors $h(i)$ larger than $h$, and such that for any $r < 1$, Alg is not $r$–robust on $\mathcal{G}$.*

We recall that, since the Byzantine nodes are unknown, given a communication network and an algorithm Alg, the honest nodes follow Alg independently of the position of Byzantine nodes in the network. Thus, it is only when the position of the Byzantine nodes satisfies some hypothesis that Alg can be r-robust.

Here we consider the hypothesis $H = \{\mu_2(\mathcal{G}_\mathcal{H}) = 2b\} \cap \{\max_{i\in\mathcal{H}} b(i) \leq b\}$. We show that for any algorithm and any $h \geq 0$, there exists a communication network $\mathcal{N}$ on which any algorithm Alg is not r-robust for (at least) one configuration of the Byzantines that verifies the hypothesis $H$.

The structure of the proof is thus the following:

1. We introduce a family of unweighted communication networks $\{\mathcal{N}_{m,k}\}_{m\in\mathbb{N},k\in[m]}$, such that $\mathcal{N}_{m,k}$ admits three symmetric configurations of Byzantines nodes with $2m$ honest nodes and $m$ Byzantines nodes, and each honest node is neighbor to $k$ Byzantine nodes and $m + k$ honest nodes. Then we show that an algorithm can not be $r$-robust in these three configurations with $r < 1$.

2. Now, for $c \geq 0$, let's denote $\mathcal{G}_{m,k,c}$ the weighted graph with a uniform weight $c$ on the edges and such that $\mathcal{G}_{m,k,c}$ is a weighted version of one of the three above-mentioned configurations of $\mathcal{N}_{m,k}$. We show that:

   (a) The weight of Byzantines in the neighborhood of any honest node is equal to $b = ck$.
   (b) The algebraic connectivity of the honest subgraph $[\mathcal{G}_{m,k,c}]_\mathcal{H}$ verifies:

   $$\mu_2([\mathcal{G}_{m,k,c}]_\mathcal{H}) = 2ck = 2b.$$

   (c) For any $h \geq b$, there exists a graph within $\{\mathcal{G}_{m,k,c}\}_{m\in\mathbb{N},k\in[m],c\in\mathbb{R}_+}$ such that all honest nodes have a weight associated to honest neighbors $h(i)$ larger than $h$.

This concludes the proof.

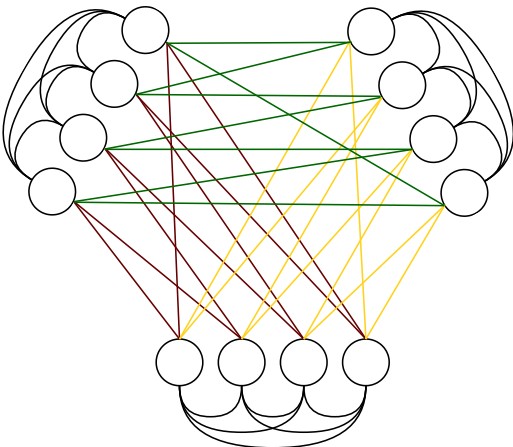

*Figure 8.* Topology of $\mathcal{N}_{m=4,k=2}$.

### E.1. Definition of the family of graphs

Let $m \in \mathbb{N}^*$ r and $k \in [m]$.

We define the communication network $\mathcal{N}_{m,k}$ as composed of three cliques of $m$ nodes: $C_1$, $C_2$, and $C_3$. Each node in $C_i$ is additionally connected to exactly $k$ nodes in $C_{i+1 \mod 3}$ and to $k$ nodes in $C_{i-1 \mod 3}$. Moreover, those connections are assumed to be *in circular order*, i.e., for any $j \in [m]$, node $j$ in $C_i$ is connected to nodes $j, \ldots, j + k \mod m$ in $C_{i+1 \mod 3}$.

We assume that one of the cliques is composed of Byzantine nodes, each honest node having $k$ Byzantine neighbors, and there are $m$ Byzantine nodes among the $3m$ nodes.

### E.2. No algorithm can be $\alpha$-robust on $\mathcal{N}_{m,k}$.

We now show that no algorithm can be robust on the communication network $\mathcal{N}_{m,k}$.

**Lemma E.2.** *Let one unknown clique within $C_1$, $C_2$, and $C_3$ be composed of Byzantine nodes, then no communication algorithm is $\alpha$-robust on $\mathcal{N}_{m,k}$.*

Assume an $r$–robust algorithm exists on the communication network $\mathcal{G}_{H,k}$. Informally:

1. We first show that if all nodes within one clique hold a unique parameter $\boldsymbol{x}$, and receive this parameter from nodes of either of the two other cliques, then they cannot change their parameter.
2. We then consider a setting where the two honest cliques hold different parameters, and we conclude that Byzantine nodes can force all honest nodes to keep their initial parameter at all times. This shows that in the considered setting, $r < 1$ is impossible.

*Proof.*
**Part I.** Let Alg be an algorithm $r$–robust on $\mathcal{N}_{m,k}$ with $r < 1$. We denote $\boldsymbol{x}_i^+$ the output from node $i$ after running Alg.

We know that one of the cliques say $C_1$, is composed of honest nodes. Let the honest nodes within $C_1$ hold the parameter $\boldsymbol{x}$. Nodes in another clique, say $C_2$, declare the parameter $\boldsymbol{x}$ as well, while nodes in $C_3$ declare another parameter, say $\boldsymbol{y} \neq \boldsymbol{x}$. We show that all nodes $i \in C_1$ must output the parameter $\hat{\boldsymbol{x}}_i = \boldsymbol{x}$.

From the point of view of nodes in $C_1$ considering that the Byzantine clique is unknown, it is impossible to distinguish between these situations:

1. Situation I: $C_2$ is honest, and $C_3$ is Byzantine,
2. Situation II: $C_2$ is Byzantine, and $C_3$ is honest.

Thus the outputs of nodes in $C_1$ after running Alg, denoted $(\boldsymbol{x}_i^+)_{i \in C_1}$, are the same in both situations.

In Situation I, nodes of $C_2$ are honest, and nodes in $C_1$ and $C_2$ have the same initial parameter; hence, the initial quadratic error is 0. The $r$ criterion writes

$$\sum_{i \in \mathcal{H}} \|\boldsymbol{x}_i^+ - \overline{\boldsymbol{x}}_{\mathcal{H}}\|^2 \leq r \sum_{i \in \mathcal{H}} \|\boldsymbol{x}_i - \overline{\boldsymbol{x}}_{\mathcal{H}}\|^2 = 0.$$

It follows that for any node $i$ in $C_1$, $\boldsymbol{x}_i^+ = \boldsymbol{x}$, *i.e.*, nodes in $C_1$ do not change their parameters (in both Situation I and Situation II).

**Part II.** Consider the setting where $C_1$ and $C_2$ are honest, while $C_3$ is Byzantine, and that nodes $C_1$ hold the parameter $\boldsymbol{x}$, while nodes in $C_2$ hold the parameter $\boldsymbol{y} \neq \boldsymbol{x}$.

As Byzantine nodes can declare different values to their different neighbors, nodes in $C_3$ can declare to nodes in $C_1$ that they hold the value $\boldsymbol{x}$, and to nodes in $C_2$ that they hold the value $\boldsymbol{y}$. Following Part I, nodes in $C_1$ and in $C_2$ cannot update their parameter, (i.e. $\forall i \in \mathcal{H}$, $\boldsymbol{x}_i^+ = \boldsymbol{x}_i$). Applying the $r$–robustness property brings:

$$\sum_{i \in \mathcal{H}} \|\boldsymbol{x}_i^+ - \overline{\boldsymbol{x}}_{\mathcal{H}}\|^2 \leq r \sum_{i \in \mathcal{H}} \|\boldsymbol{x}_i - \overline{\boldsymbol{x}}_{\mathcal{H}}\|^2 = r \sum_{i \in \mathcal{H}} \|\boldsymbol{x}_i^+ - \overline{\boldsymbol{x}}_{\mathcal{H}}\|^2,$$

which implies that $r \geq 1$ since $\sum_{i \in \mathcal{H}} \|\boldsymbol{x}_i^+ - \overline{\boldsymbol{x}}_{\mathcal{H}}\|^2 > 0$. It follows that the Algorithm Alg is not $r$–robust on $\mathcal{N}_{m,k}$ for an $r < 1$.

$\square$

### E.3. Spectral properties of the graph.

We define as $\mathcal{G}_{m,k,c}$ the graph associated with the network $\mathcal{N}_{m,k}$ where all edges have weight $c$ and the nodes in one of the three cliques are Byzantine.

We show here that $\mu_2([\mathcal{G}_{m,k,c}]_{\mathcal{H}}) = 2kc$. This concludes the proof as we have:

1. The weight of Byzantines in the neighborhood of any honest node is equal to $b = ck$;

2. Which is linked to the algebraic connectivity of the honest subgraph $\mu_2([\mathcal{G}_{m,k,c}]_{\mathcal{H}}) = c2k$;

3. While the weight of honest nodes in the neighborhood of honest nodes is equal to $h = c(m + k)$, thus growths to infinity with $m$;

**Analysis.** We denote $\boldsymbol{L}_{\mathcal{H}}$ the Laplacian of $[\mathcal{G}_{m,k,c=1}]_{\mathcal{H}}$, the honest subgraph of the network $\mathcal{N}_{m,k}$ in which we provided unitary weights to edges. We show that $\mu_2([\mathcal{G}_{m,k,c=1}]_{\mathcal{H}}) = 2k$, which brings that $\mu_2([\mathcal{G}_{m,k,c}]_{\mathcal{H}}) = 2kc$.

**Lemma E.3.** *The second smallest eigenvalue of the (unitary weighted) Laplacian matrix $\boldsymbol{L}_{\mathcal{H}}$ of $[\mathcal{G}_{m,k,1}]_{\mathcal{H}}$ is equal to $\mu_2(\boldsymbol{L}_{\mathcal{H}}) = 2k$.*

*Proof.* Recall that $m \triangleq m$. Let $\boldsymbol{M} \in \mathbb{R}^{m \times m}$ be a circulant matrix defined as $\boldsymbol{M} = \sum_{q=0}^{k-1} \boldsymbol{J}^q$, where $\boldsymbol{J}$ denotes the permutation

$$\boldsymbol{J} := \begin{pmatrix} 0 & 1 & 0 & \ldots & 0 \\ 0 & 0 & 1 & \ldots & 0 \\ \vdots & & & \ddots & \vdots \\ 0 & & & & 1 \\ 1 & 0 & 0 & \ldots & 0 \end{pmatrix}.$$

The Laplacian matrix of the honest subgraph of $\mathcal{G}_{H,k}$ can be written as:

$$\boldsymbol{L}_{\mathcal{H}} = \begin{pmatrix} m\boldsymbol{I}_m - \boldsymbol{1}_m\boldsymbol{1}_m^T & 0 \\ 0 & m\boldsymbol{I}_m - \boldsymbol{1}_m\boldsymbol{1}_m^T \end{pmatrix} + \begin{pmatrix} k\boldsymbol{I}_m & -\boldsymbol{M} \\ -\boldsymbol{M}^T & k\boldsymbol{I}_m \end{pmatrix}$$

Hence

$$\boldsymbol{L}_{\mathcal{H}} = (k+m)\boldsymbol{I}_{2m} - \begin{pmatrix} \mathbf{1}_m\mathbf{1}_m^T & 0 \\ 0 & \mathbf{1}_m\mathbf{1}_m^T \end{pmatrix} - \begin{pmatrix} 0 & \boldsymbol{M} \\ \boldsymbol{M}^T & 0 \end{pmatrix}. \tag{10}$$

This matrix decomposition allows to have the eigenvalues of the the matrix $W_{\mathcal{G}}$.

**Lemma E.4.** *The eigenvalues of $\boldsymbol{L}_{\mathcal{H}}$ are $\{0, 2k\} \cup \{k + m \pm |\sum_{q=0}^{k-1} \omega^{pq}|; p \in \{1, \ldots, m-1\}\}$ where $\omega := \exp(\frac{2i\pi}{m})$.*

To prove the Lemma E.4, we first need to following result.

**Lemma E.5.** *If $\boldsymbol{A}$ is a symmetric matrix in $\mathbb{R}^{2m \times 2m}$, which can be decomposed as $\boldsymbol{A} = \begin{pmatrix} 0 & \boldsymbol{M} \\ \boldsymbol{M}^T & 0 \end{pmatrix}$, where $\boldsymbol{M} \in \mathbb{R}^{m \times m}$ is a matrix with complex eigenvalues $\mu_0, \ldots, \mu_{m-1}$.*

*Then the eigenvalues of $\boldsymbol{A}$ are $\{\pm|\mu_p|; p = 0, \ldots, m-1\}$.*

*Proof of Lemma E.5.* Let $\boldsymbol{M} = \boldsymbol{U}^*\boldsymbol{D}\boldsymbol{U}$ be the diagonalization of $\boldsymbol{M}$ where $\boldsymbol{D} = \mathrm{Diag}(\mu_0, \ldots, \mu_{m-1})$, and $\boldsymbol{U}$ is a unitary matrix, i.e. $\boldsymbol{U}\boldsymbol{U}^* = \boldsymbol{I}$ where we denote as $\boldsymbol{U}^* = \overline{\boldsymbol{U}}^T$ the conjugate transpose of $\boldsymbol{U}$, where the (simple) conjugate matrix is denoted $\overline{\boldsymbol{U}}$.

Lemma E.5 follows from

$$\begin{pmatrix} 0 & \boldsymbol{M} \\ \boldsymbol{M}^T & 0 \end{pmatrix} = \begin{pmatrix} 0 & \boldsymbol{U}^*\boldsymbol{D}\boldsymbol{U} \\ \boldsymbol{U}^T\boldsymbol{D}\overline{\boldsymbol{U}} & 0 \end{pmatrix} \underset{\overline{\boldsymbol{A}}=\boldsymbol{A}}{=} \begin{pmatrix} 0 & \boldsymbol{U}^*\boldsymbol{D}\boldsymbol{U} \\ \boldsymbol{U}^*\overline{\boldsymbol{D}}\boldsymbol{U} & 0 \end{pmatrix}.$$

Hence

$$\boldsymbol{A} = \begin{pmatrix} \boldsymbol{U}^* & 0 \\ 0 & \boldsymbol{U}^* \end{pmatrix} \begin{pmatrix} 0 & \boldsymbol{D} \\ \overline{\boldsymbol{D}} & 0 \end{pmatrix} \begin{pmatrix} \boldsymbol{U} & 0 \\ 0 & \boldsymbol{U} \end{pmatrix}.$$

A simple calculus (using that $\boldsymbol{D}$ is diagonal) yields that all eigenvalues of $\begin{pmatrix} 0 & \boldsymbol{D} \\ \overline{\boldsymbol{D}} & 0 \end{pmatrix}$ are $\{\pm|\boldsymbol{D}_p|; p = 0, \ldots, m-1\}$. $\square$

*Proof of Lemma E.4.* We start from the decomposition of Equation (10) :

$$\boldsymbol{L}_{\mathcal{H}} = (k+m)\boldsymbol{I}_{2m} - \begin{pmatrix} \mathbf{1}_m\mathbf{1}_m^T & 0 \\ 0 & \mathbf{1}_m\mathbf{1}_m^T \end{pmatrix} - \begin{pmatrix} 0 & \boldsymbol{M} \\ \boldsymbol{M}^T & 0 \end{pmatrix}.$$

We first notice that the subspace spanned by $(\mathbf{1}_m^T, +\mathbf{1}_m^T)^T$ and $(\mathbf{1}_m^T, -\mathbf{1}_m^T)^T$ is an eigenspace of $\begin{pmatrix} \mathbf{1}_m\mathbf{1}_m^T & 0 \\ 0 & \mathbf{1}_m\mathbf{1}_m^T \end{pmatrix}$ associated the eigenvalue $m$, and the orthogonal subspace is associated with 0. Furthermore these are eigenvectors of $\begin{pmatrix} 0 & \boldsymbol{M} \\ \boldsymbol{M}^T & 0 \end{pmatrix}$ associated with $k$ and $-k$. It follows that they are eigenvectors of $\boldsymbol{L}_{\mathcal{H}}$ with eigenvalues 0 and $2k$. We notice as well that the three matrices of Equation (10) can be diagonalized in the same orthogonal basis.

The matrix $\boldsymbol{M}$ is a circulant matrix, so it can be diagonalized in $\mathbb{C}$. The eigenvalues are $\{\mu_p = \sum_{q=0}^{k-1} \omega^{pq}; p \in \{0, \ldots, m-1\}\}$, where $\omega := \exp(\frac{2i\pi}{m})$. The eigenvector associated with $\mu_p$ is $x_p = (1, \omega^p, \ldots, \omega^{(m-1)p})^T$. As such, with $\boldsymbol{U} = (x_0, \ldots, x_{m-1})$ and $\boldsymbol{D} = \mathrm{Diag}(\mu_0, \ldots, \mu_{m-1})$, $\boldsymbol{M}$ writes:

$$\boldsymbol{M} = \boldsymbol{U}^*\boldsymbol{D}\boldsymbol{U}.$$

Considering Lemma E.5, the eigenvalue of $\begin{pmatrix} 0 & \boldsymbol{M} \\ \boldsymbol{M}^T & 0 \end{pmatrix}$ are $\{\pm|\mu_p|; p = 0, \ldots, m-1\}$, considering that $p = 0$ corresponds to the eigenvalues $+k$ and $-k$, hence the eigenvectors $(\mathbf{1}_m^T, \mathbf{1}_m^T)^T$ and $(\mathbf{1}_m^T, -\mathbf{1}_m^T)^T$, we deduce that the eigenvalues of $\boldsymbol{L}_{\mathcal{H}}$ are $\{\pm|\mu_p|; p = 0 \ldots m-1\}$. $\square$

**End of the proof of Lemma E.3.**

To prove Lemma E.3, considering the decomposition of Equation (10), we only have to show that $m - k$ is always the second largest eigenvalue of the matrix

$$\boldsymbol{B} := \begin{pmatrix} \mathbf{1}_m \mathbf{1}_m^T & 0 \\ 0 & \mathbf{1}_m \mathbf{1}_m^T \end{pmatrix} + \begin{pmatrix} 0 & \boldsymbol{M} \\ \boldsymbol{M}^T & 0 \end{pmatrix}.$$

First, considering Lemma E.4, the eigenvalues of $B$ are $\{m+k, m-k\} \cup \{\pm|\mu_p|; p \in \{1, \ldots, m-1\}\}$ with $\mu_p = \sum_{q=0}^{k-1} \omega^{pq}$. As such showing that $|\mu_p| \leq m - k$ if $p \in \{1, \ldots, m-1\}$ yields the result.

As $\omega^{mp} = \omega^{0p} = 1$, we have that $\sum_{q=0}^{m-1} \omega^{pq}(1 - \omega^p) = 0$. Hence, for $p \in \{1, \ldots, m-1\}$, as $\omega^p \neq 1$,

$$\sum_{q=0}^{m-1} \omega^{pq} = 0 \implies \mu_p = \sum_{q=0}^{k-1} \omega^{pq} = -\sum_{q=k}^{m-1} \omega^{pq}.$$

It follows from $|\omega| = 1$ that for $p \in \{1, \ldots, m-1\}$, $|\mu_p| \leq m - k$. $\qquad\square$

# F. $(b, \rho)$-robust summation rules

We recall the definition of summation rules.

**Definition F.1** ($(b, \rho)$–robust summation). Let $b, \rho \geq 0$. An aggregation rule $F : (\mathbb{R}_+ \times \mathbb{R}^d)^n \to \mathbb{R}^d$ is a $(b, \rho)$–robust summation, when, for any vectors $(\boldsymbol{z}_i)_{i \in [n]} \in (\mathbb{R}^d)^n$, any weights $(\omega_i)_{i \in [n]} \in \mathbb{R}_+^n$ and any set $S \subset [n]$ such that $\sum_{i \in \overline{S}} \omega_i \leq b$, there is

$$\left\| F\big((\omega_i, \boldsymbol{z}_i)_{i \in [n]}\big) - \sum_{i \in S} \omega_i \boldsymbol{z}_i \right\|^2 \leq \rho b \sum_{i \in S} \omega_i \|\boldsymbol{z}_i\|^2.$$

where $\overline{S} := [n] \backslash S$.

## F.1. Clipping

We first prove the robustness of the clipping-based summation.

**Proposition F.2.** *Let $b \geq 0$, then*

1. *(Practical)* $\mathrm{CS}_+$ *is $(b, \rho)$–robust with $\rho = 2$.*
2. *(Oracle)* $\mathrm{CS}_+^{or.}$ *is $(b, \rho)$–robust with $\rho = 1$.*
3. *(Oracle)* $\mathrm{CS}_{He}^{or.}$ *is $(b, \rho)$–robust with $\rho = 4$.*

*Proof.* Let $(\omega_i, \boldsymbol{z}_i)_{i \in [n]} \in (\mathbb{R}_+ \times \mathbb{R}^d)^n$, and $S \subset [n]$ such that $\sum_{i \in \overline{S}} \omega_i \leq b$,

We consider for $\tau \geq 0$

$$F\big((\omega_i, \boldsymbol{z}_i)_{i=1,\ldots,n}\big) := \sum_{i=1}^n \omega_i \operatorname{Clip}(\boldsymbol{z}_i; \tau).$$

Then, by applying the triangle inequality we have

$$\left\| F\big((\omega_i, \boldsymbol{z}_i)_{i=1,\ldots,n}\big) - \sum_{i \in S} \omega_i \boldsymbol{z}_i \right\|^2 = \left\| \sum_{i \in S} \omega_i \left( \operatorname{Clip}(\boldsymbol{z}_i; \tau) - \boldsymbol{z}_i \right) + \sum_{i \in \overline{S}} \omega_i \operatorname{Clip}(\boldsymbol{z}_i; \tau) \right\|^2$$

$$\leq \left( \sum_{i \in S} \omega_i \|\boldsymbol{z}_i - \operatorname{Clip}(\boldsymbol{z}_i; \tau)\| + \sum_{i \in \overline{S}} \omega_i \| \operatorname{Clip}(\boldsymbol{z}_i; \tau)\| \right)^2$$

$$\leq \left( \sum_{i \in S} \omega_i (\|\boldsymbol{z}_i\| - \tau)_+ + b\tau \right)^2. \tag{11}$$

Where we used $\sum_{i \in \overline{S}} \omega_i \leq b$.

**1. Case of our clipping threshold.** We choose $\tau$ as

$$\tau = \max \left\{ \tau \geq 0 : \sum_{i=1}^{n} \omega_i \mathbf{1}_{\|\boldsymbol{z}_i\| \geq \tau} \geq 2b \right\} \triangleq \tau_{\text{ours}}\big((\omega_i, \boldsymbol{z}_i)_{i \in [n]}\big).$$

This corresponds to lowering the clipping threshold until the sum of the weights of clipped vectors is essentially equal to $2b$. This ensures that the total weight of the honest vectors that are clipped falls between $b$ and $2b$. If there are ties at the clipping threshold, honest vectors can be arbitrarily denoted as *clipped* or *non-clipped*. Indeed, there is no clipping error incurred since the clipping threshold is the same as the actual value of the difference. Therefore, we do not have to accumulate error for the weight over $2b$, so the following equation always holds:

$$2b \geq \sum_{i \in S} \omega_i \mathbf{1}_{i \text{ clipped}} \geq b.$$

Which allows us to write:

$$\sum_{i \in S} \omega_i \left( \|\boldsymbol{z}_i\| - \tau \right)_+ + b\tau \leq \sum_{i \in S} \omega_i (\|\boldsymbol{z}_i\| - \tau) \mathbf{1}_{i \text{ clipped}} + b\tau$$

$$\leq \sum_{i \in S} \omega_i \|\boldsymbol{z}_i\| \mathbf{1}_{i \text{ clipped}}.$$

We conclude the proof using the Cauchy-Schwarz inequality:

$$\left\| F\big((\omega_i, \boldsymbol{z}_i)_{i=1,\dots,n}\big) - \sum_{i \in S} \omega_i \boldsymbol{z}_i \right\|^2 \leq \left( \sum_{i \in S} \sqrt{\omega_i} \|\boldsymbol{z}_i\| \cdot \sqrt{\omega_i} \mathbf{1}_{i \text{ clipped}} \right)^2$$

$$\leq \left( \sum_{i \in S} \omega_i \mathbf{1}_{i \text{ clipped}} \right) \left( \sum_{i \in S} \omega_i \|\boldsymbol{z}_i\|^2 \right)$$

$$\leq 2b \sum_{i \in S} \omega_i \|\boldsymbol{z}_i\|^2.$$

Where we used $\sum_{j \in n_{\mathcal{H}}(i)} \omega_{ij} \mathbf{1}_{j \text{ clipped}} \leq 2b$. Note that, if somehow it is possible to choose the clipping threshold such that the weight of clipped vectors within $S$ is equal exactly to $b$ then this factor 2 disappears. Which correspond to our oracle clipping threshold $\tau_{\text{ours}}^{\text{or.}}$. The same result can be achieved when it is possible to identify a subset of $\{1, \dots, n\}$ of weight $2b$ which includes the set $\overline{S}$, and if the weight with $\overline{S}$ sums exactly to $b$. This is for instance the case of the communication graph used in Appendix E: nodes in (for instance) $C_1$ know that Byzantines nodes are among the nodes within $C_2$ and $C_3$, thus selecting all their neighbors that belong to these two other cliques leads to a subset of neighbors of weight $2b$ with exactly a weight $b$ corresponding to Byzantine neighbors.

**2. Clipping threshold of He et al. (2023)** We plug in Equation (11) the following upper bound:

$$(\|\boldsymbol{z}_i\| - \tau)_+ = \tau \left( \frac{\|\boldsymbol{z}_i\|}{\tau} - 1 \right)_+ \leq \tau \left( \frac{\|\boldsymbol{z}_i\|^2}{\tau^2} - 1 \right)_+ \leq \frac{\|\boldsymbol{z}_i\|^2}{\tau}.$$

This yields

$$\left\| F\big((\omega_i, \boldsymbol{z}_i)_{i=1,\dots,n}\big) - \sum_{i \in S} \omega_i \boldsymbol{z}_i \right\|^2 \leq \left( \sum_{i \in S} \omega_i \frac{\|\boldsymbol{z}_i\|^2}{\tau} + b\tau \right)^2.$$

Then taking as clipping threshold the minimizer of the RHS, $\tau^* = \sqrt{\frac{1}{b} \sum_{i \in S} \omega_i \|\boldsymbol{z}_i\|^2} \triangleq \tau_{\text{He}}^{\text{or.}}\big((\omega_i, \boldsymbol{z}_i)_{i \in [n]}\big)$ leads to:

$$\left\| F\big((\omega_i, \boldsymbol{z}_i)_{i=1,\dots,n}\big) - \sum_{i \in S} \omega_i \boldsymbol{z}_i \right\|^2 \leq 4b \sum_{i \in S} \omega_i \|\boldsymbol{z}_i\|^2.$$

However, the clipping threshold here requires an exact knowledge of the set $S$. Furthermore, it is unclear to what extend making an approximate estimate of this clipping threshold allows us to derive robustness guarantees. □

*Remark* F.3. A key point here is that this oracle clipping threshold corresponds to the **unique minimizer** within each squared term of the sum. Hence, considering for instance the adaptive practical clipping rule of (He et al., 2023) leads to a **larger upper bound on the error**.

### F.2. Geometric trimming a.k.a. NNA

Let $(\omega_i, \boldsymbol{z}_i)_{i \in [n]} \in (\mathbb{R}_+ \times \mathbb{R}^d)^n$, and $S \subset [n]$ such that $\sum_{i \in \overline{S}} \omega_i \leq b$,

We recall the definition of GTS: Assume w.l.o.g. that $(\|\boldsymbol{z}_i\|)_{i \in [n]}$ are sorted, i.e. $\|\boldsymbol{z}_1\| \leq \ldots \leq \|\boldsymbol{z}_n\|$, and denote $k^*(b) := \max\{k \in [n]; \sum_{i \geq k} \omega_i \geq b\}$ the index of the largest vector which has at least a weight $b$ of vector largest than him. (GTS) computes $\tilde{\omega}_{k^*(b)} := \sum_{i \geq k^*(b)} \omega_i - b$, and outputs

$$\text{GTS}\big((\omega_i, \boldsymbol{z}_i)_{i \in [n]}\big) = \tilde{\omega}_{k^*(b)} \boldsymbol{z}_{k^*} + \sum_{i < k^*(b)} \omega_i \boldsymbol{z}_i.$$

**Lemma F.4.** *Geometric trimming is* $(b, \rho)$ *-robust with* $\rho = 4$.

*Proof.* Let $(\omega_i, \boldsymbol{z}_i)_{i \in [n]} \in (\mathbb{R}_+ \times \mathbb{R}^d)^n$, and $S \subset [n]$ such that $\sum_{i \in \overline{S}} \omega_i \leq b$, Without loss of generality we assume that $\tilde{\omega}_{k^*(b)} = 0$[4].

Thus the aggregation rules write

$$F\big((\omega_i, \boldsymbol{z}_i)_{i=1,\ldots,n}\big) := \sum_{i < k^*(b)} \omega_i \boldsymbol{z}_i. = \sum_{i=1}^{n} \omega_i \boldsymbol{z}_i \mathbf{1}_{i \text{ not removed}}$$

Then, by applying the triangle inequality we have

$$\left\| F\big((\omega_i, \boldsymbol{z}_i)_{i=1,\ldots,n}\big) - \sum_{i \in S} \omega_i \boldsymbol{z}_i \right\|^2 = \left\| \sum_{i \in S} \omega_i \boldsymbol{z}_i \mathbf{1}_{i \text{ removed}} + \sum_{i \in \overline{S}} \omega_i \boldsymbol{z}_i \mathbf{1}_{i \text{ not removed}} \right\|^2$$

$$\leq \left( \sum_{i \in S} \omega_i \|\boldsymbol{z}_i\| \mathbf{1}_{i \text{ removed}} + \sum_{i \in \overline{S}} \omega_i \|\boldsymbol{z}_i\| \mathbf{1}_{i \text{ not removed}} \right)^2.$$

As $\sum_{i=1}^{n} \omega_i \mathbf{1}_{i \text{ removed}} = b$, it holds that

$$\sum_{i \in \overline{S}} \omega_i \leq b = \sum_{i=1}^{n} \omega_i \mathbf{1}_{i \text{ removed}} = \sum_{i \in S} \omega_i \mathbf{1}_{i \text{ removed}} + \sum_{i \in \overline{S}} \omega_i \mathbf{1}_{i \text{ removed}}$$

$$\implies \sum_{i \in \overline{S}} \omega_i \mathbf{1}_{i \text{ not removed}} \leq \sum_{i \in S} \omega_i \mathbf{1}_{i \text{ removed}}.$$

Furthermore, if $i$ is removed and $j$ is not, then $\|\boldsymbol{z}_i\| \geq \|\boldsymbol{z}_j\|$. It follows that

$$\sum_{i \in S} \omega_i \|\boldsymbol{z}_i\| \mathbf{1}_{i \text{ removed}} \geq \sum_{i \in \overline{S}} \omega_i \|\boldsymbol{z}_i\| \mathbf{1}_{i \text{ not removed}}.$$

---

[4]Which can be ensured by adding artificially the entry $(\tilde{\omega}_{k^*(b)}, \boldsymbol{z}_{k^*(b)})$.

Consequently:

$$\left\| F\big((\omega_i, \boldsymbol{z}_i)_{i=1,\dots,n}\big) - \sum_{i \in S} \omega_i \boldsymbol{z}_i \right\|^2 \le \left( 2 \sum_{i \in S} \omega_i \|\boldsymbol{z}_i\| \mathbf{1}_{i\text{ removed}} \right)^2$$

$$= 4 \left( \sum_{i \in S} \sqrt{\omega_i} \|\boldsymbol{z}_i\| \sqrt{\omega_i} \mathbf{1}_{i\text{ removed}} \right)^2$$

$$\le 4 \sum_{i \in S} \omega_i \mathbf{1}_{i\text{ removed}} \sum_{i \in S} \omega_i \|\boldsymbol{z}_i\|^2 \qquad \text{using Cauchy-Schwarz.}$$

Thus:

$$\left\| F\big((\omega_i, \boldsymbol{z}_i)_{i=1,\dots,n}\big) - \sum_{i \in S} \omega_i \boldsymbol{z}_i \right\|^2 \le 4b \sum_{i \in S} \omega_i \|\boldsymbol{z}_i\|^2.$$

$\square$

# G. Proofs for D-SGD

*Proof of Corollary 4.8.* This proof hinges on the fact that the proof of Farhadkhani et al. (2023, Theorem 1) does not actually require that communication is performed using NNA, but simply that the aggregation procedure respects $(\alpha, \lambda)$-reduction, which they prove in their Lemma 2. Then, all subsequent results invoke this Lemma instead of the specific aggregation procedure. $\text{CS}_+$-RG also satisfies $(\alpha, \lambda)$-reduction, as we prove in Theorem 3.3. We can then use their bounds on the errors out of the box.

Then, as $T$ grows, and ignoring constant factors, only the first and last terms in their Theorem 3 remain, leading to, for $i \in \mathcal{H}$:

$$\frac{1}{T} \sum_{t=1}^{T} \mathbb{E}\left[ \|\nabla f_{\mathcal{H}}(\boldsymbol{x}_i^t)\|^2 \right] = \mathcal{O}\left( L\sigma \sqrt{\frac{f(\overline{\boldsymbol{x}}_{\mathcal{H}}^0) - f(\boldsymbol{x}^*)}{T}}(|\mathcal{H}|^{-1} + C) + C\zeta^2 \right), \tag{12}$$

where $C = c_1 + \lambda + \lambda c_1$, with $c_1 = \alpha(1+\alpha)/(1-\alpha)^2$. Note that we give $\mathcal{O}()$ versions of the Theorems for simplicity, but Farhadkhani et al. (2023, Theorem 1) allows to derive precise upper bounds for any $T \ge 1$.

**One-step derivation.** The one-step result is obtained by taking the values of $\alpha = 1 - \gamma(1-\delta)$ and $\lambda = \gamma\delta$. On the one hand

$$c_1 = \frac{(1-\gamma(1-\delta))(2-\gamma(1-\delta))}{\gamma^2(1-\delta)^2} = O\left( \frac{1-\gamma(1-\delta)}{\gamma^2(1-\delta)^2} \right).$$

On the other hand, $\lambda = \gamma\delta \le 1$ implies that $C = O(c_1)$. Both lead to

$$\frac{1}{T} \sum_{t=1}^{T} \mathbb{E}\left[ \|\nabla f_{\mathcal{H}}(\boldsymbol{x}_i^t)\|^2 \right] = \mathcal{O}\left( L\sigma \sqrt{\frac{f(\overline{\boldsymbol{x}}_{\mathcal{H}}^0) - f(\boldsymbol{x}^*)}{T}}\left(|\mathcal{H}|^{-1} + \frac{1-\gamma(1-\delta)}{\gamma^2(1-\delta)^2}\right) + \frac{1-\gamma(1-\delta)}{\gamma^2(1-\delta)^2}\zeta^2 \right), \tag{13}$$

The result of Corollary 4.8 derives from the case $\gamma \ll 1$ (otherwise, the guarantees are essentially the same as in Farhadkhani et al. (2023)), and $\delta \ge \frac{1}{\mathcal{H}}$ (otherwise there is essentially no Byzantine).

**Multi-step derivation.** In the previous case, we see that $C$ is dominated by the $c_1$ term since $c_1 >> \lambda$. In particular, the guarantees would increase if we were able to trade-off some $\alpha$ for some $\lambda$, which is possible by using multiple communications steps. This is what we do, and take enough steps that $c_1 = o(\lambda)$ (i.e., $\alpha \approx 0$), so that $C \approx \lambda$. Following Corollary 3.4, it can be achieved by performing $\tilde{O}(\gamma^{-1}(1-\delta)^{-1})$ steps of F-RG, where logarithmic factors are hidden in the $\tilde{O}$ notation. We then plug the multi-step $\lambda$ value from Corollary 3.4 to obtain the result: as $\lambda = O\left(\frac{\delta}{\gamma(1-\delta)}\right)$, the multi-communication steps convergence bound writes

$$\frac{1}{T} \sum_{t=1}^{T} \mathbb{E}\left[ \|\nabla f_{\mathcal{H}}(\boldsymbol{x}_i^t)\|^2 \right] = \mathcal{O}\left( L\sigma \sqrt{\frac{f(\overline{\boldsymbol{x}}_{\mathcal{H}}^0) - f(\boldsymbol{x}^*)}{T}}\left(|\mathcal{H}|^{-1} + \frac{\delta}{\gamma(1-\delta)^2}\right) + \frac{\delta}{\gamma(1-\delta)^2}\zeta^2 \right). \tag{14}$$

□

