# OpenReview forum: "Unified Breakdown Analysis for Byzantine Robust Gossip"
_ICML.cc/2025/Conference — ICML 2025 poster_

### Official Review · Reviewer_EynP · 2025-03-11

**Overall Recommendation:** 4

**Summary:**

This paper addresses the problem of designing robust decentralized algorithms in the face of so-called Byzantine adversaries, i.e. adversaries likely to send arbitrary (and potentially equivocal) information to other participants during protocol execution. It introduces a general framework, F-RG, for robust decentralized communication algorithms to compute weighted sum queries over sparse communication graphs. It provides a theoretical analysis of the breakdown point of this framework, presenting tight upper bounds on the number (maximal weight) of adversaries a system can tolerate with respect to the spectral properties of the communication graph. The algorithms matching this bound are, however, impractical. Accordingly, the paper presents an aggregation rule, $CS_{ours}$, that achieves near-optimal performance at the breakdown point. Combined with the existing mixing conditions of robust decentralized optimization, F-RG provides a fully robust pipeline to robustly implement decentralized SGD (with associated convergence analysis under standard assumptions). The paper also introduces a new adversarial strategy that exploits the spectral properties of the communication graph to disrupt decentralized learning. Empirical experiments are conducted to validate the theoretical results, comparing $CS_{ours}$ with other state-of-the-art aggregation schemes in various attack scenarios for summation and decentralized learning problems.

**Claims And Evidence:**

The paper claims that F-RG provides a unified framework for Byzantine-robust decentralized summation and that $CS_{ours}$ achieves a near-optimal breakdown point. These claims are supported by:

- The introduction of the $(b, \rho)$-robustness criterion, which generalizes robustness guarantees for weighted summation queries to arbitrary sparse networks.
- Theoretical upper and lower bounds demonstrating that, in the presented framework, the breakdown point is tightly characterized by the condition $\mu_{\text{min}} < 2 b $.
- Theoretical results demonstrating that $CS_{ours}$ is robust as soon as $\mu_{\text{min}} \geq 4b $, hence demonstrating the near-optimality of $CS_{ours}$ in regard to the breakdown point.
- Extension of the robust summation guarantees to robust Decentralized SGD using the notion of $(\alpha, \lambda)$-reduction from (Farhadkhani 2022).
- Empirical results showing that $CS_{ours}$ outperforms existing methods such as NNA and ClippedGossip under the new Spectral Heterogeneity attack.

**Essential References Not Discussed:**

The paper covers the relevant literature.

**Experimental Designs Or Analyses:**

Experiments, while rather limited in terms of scale are well-presented, effectively demonstrate the robustness of the proposed aggregation rule under Byzantine attacks in sparse networks. However, the study considers only one sparse graph. I think it would be valuable to test different levels of graph connectivity to see if all methods collapse to the same performance in fully connected settings or if certain approaches are more robust across different topologies.

**Methods And Evaluation Criteria:**

The method and evaluation criteria are well suited to the claims of the paper and to the problem being tackled. The technical contribution is clear and well-explained, and the experimental setup is clearly described.

**Other Comments Or Suggestions:**

- I believe that Section 3.4 should be titled "Fundamental limits of $r$-robustness" to better reflect the fact that the lower bound applies only to $r$-robust communication schemes, as noted in Remark 3.6.

A minor technical clarification:
- In Remark 3.2, the claim that $(f, \kappa)$-robustness can only provide $(b, \rho)$-robust summation with constant weights seem too restrictive to me. As I understand it, it seems possible to apply an $(f, \kappa)$-robust function $F$ to a transformed set of vectors $ ( (n-f) w_i z_i )_{i \in [n]}$, which would yield $(b, \rho)$-robust summation with $\rho = \kappa$, $b = n-f$, for general weights $(w_i)_{i \in [n]}$ (as long as they are upper-bounded by 1). Maybe I missed something though, could the authors comment on this point ?

- Maybe the author could reduce a bit the space dedicated to listing the aggregations and instead present a bit more deeply the convergence framework (maybe re-presenting the definition of reduction being used).

Small typos:
- I think the oracle clippings presented in page 6 should be defined by taking the set $S$ as an entry, otherwise the aggregation seem a bit ill-defined.
- There is a sentence missing at the end of paragraph 1. In appendix A.

**Other Strengths And Weaknesses:**

NA

**Questions For Authors:**

Can you please answer to my wondering on Remark 3.2 and to my suggestion for additional experiments with several levels of connectivity?

**Relation To Broader Scientific Literature:**

This article makes a significant contribution to robust decentralized learning. Theoretical results generalize known limitations, while experimental analysis introduces new adversarial strategies. The article presents all the background work necessary for the reader to understand the whole picture.

**Theoretical Claims:**

The analysis is rigorous, proofs are well-written appear correct upon review. However, in my opinion , some aspects of the theoretical presentation could be improved:

- The introduction of new aggregation rules (starting on page 6) is somewhat rushed, with heavy notation and a list-based presentation that could benefit from more detailed explanations.
- The discussion of asynchronous communication is brief and lacks clarity for me on its practical implications. It may be useful to remove it from the main body of the paper and have a dedicated section in the appendix.
- Section 4.2 presents strong results on robust Distributed-SGD but does not clearly highlight the significance of using the reduction property from (Farhadkhani 2023) to ensure robust convergence guarantees. Reintroducing the definition of reduction could help make this clearer.

---

> ### Author Rebuttal · Authors · 2025-04-01
>
> We thank the reviewer for their comments and their precious suggestions. We examine the questions raised below.
>
> # Additional Experiments
>
> Following the reviewer's suggestion, we performed new experiments on Erdős-Rényi graphs and additionally compared ourselves with the IOS algorithm [Wu et al., 2022]. We refer the reviewer to the response to Reviewer UczM for further details.
>
> # Comment on Remark 3.2
>
> We fully agree with the reviewer that there is no major technical difficulty in either:
> i) defining $(f,\kappa)$-robustness with arbitrary weights, or
> ii) going from $(f,\kappa)$-robustness to $(b,\rho)$-robustness with arbitrary weights.
>
> In this remark, we mostly wanted to recall that $(f,\kappa)$-robustness is originally defined with constant weights, so that the reader can easily compare both definitions. Still, we do not believe that generalizing it to arbitrary weights is a major contribution.
>
> For now, if proofs on $(f,\kappa)$-robust aggregators have been written with constant weights, arbitrary weights can be achieved, for instance by increasing artificially the number of input vectors so that the frequency of each input approaches the targeted weight.
>
> # Suggestions
>
> We thank the reviewer for the meaningful and precious advice on the flow of the paper, which we will implement.
>
> 1. **Introduction of Aggregation Rules**
> We agree with the reviewer that this part takes space while not being clear enough. We will think about how to improve it. One solution we are investigating is to consider in the main part of the article definitions with unitary weights only and defer the generic definitions to the appendix. We agree that currently, the definition of $(b,\rho)$-robustness is a bit too far away from the example of robust summands.
>
> 2. **Asynchronous Part**
> We fully agree that it is a bit rushed; we will move it to the appendix to clarify it.
>
> 3. **Title of Section 3.4**
> The reviewer is right to point out that our limits only apply when we consider $r$-robustness as the robustness definition of any decentralized algorithm. Thus, we will follow their suggestion.
>
> 4. **Defining $(\alpha,\lambda)$-Reduction**
> We agree as well that defining it properly will make the paper easier to read. Furthermore, it will allow us to better compare it with $r$-robustness.
>
> 5. **Typos**
> We agree that currently, the 'oracle' clipping thresholds are ill-defined; we will improve this point. Thank you as well for pointing out the missing sentence.

---

### Official Review · Reviewer_GrPa · 2025-03-13

**Overall Recommendation:** 4

**Summary:**

Decentralized training often encounter adversaries that may cause degraded model without proper defenses. This paper considers the problem of robust decentralized training against Byzantine adversaries. This paper revisits the previous robust gossip schemes and propose a generic framework which recovers these baselines as special cases. As a result, they propose the F-RG algorithm that outperforms previous works and reach near optimal breakdown point. Besides, they also propose a novel attack that based on spectral heterogeneity that pushes honest workers away from each other. The theoretical convergence rates are given under standard assumptions and impossibility results are also given. The empirical results show the effectiveness of their proposed algorithm.

**Claims And Evidence:**

Yes.

**Essential References Not Discussed:**

No

**Experimental Designs Or Analyses:**

Yes. The experiment settings looks good to me.

**Methods And Evaluation Criteria:**

The baselines are common in the literature and therefore make sense.

**Other Comments Or Suggestions:**

No.

**Other Strengths And Weaknesses:**

### Strength

- This paper approach the problem by considering a generic framework that includes previous works and propose better robust gossip algorithms. The rates and algorithms therefore makes sense.
- Originality: the analysis framework, algorithm, and attack are all novel.
- Clarity: Mostly clear and self-contained.
- significance: the theoretical results of this paper is significant enough for a conference paper.

**Questions For Authors:**

This paper consider graphs from $\Gamma_{\mu,b}$. But in practice, there is an extra degree of freedom to choose the edge weights, e.g., using Metropolis weight. In this sense, both the spectral gap and the weights of Byzantine neighbors depend on such choice. It means we may not know a tight upper bound $b$ and if chosen graph belongs to $\Gamma_{\mu,b}$. Besides, the clipping strategy $CS_{ours}$ also depends on this information.  Could you leave some comments on how to handle this case?

**Relation To Broader Scientific Literature:**

No

**Theoretical Claims:**

The theoretical claim are made under standard assumptions in the literature.

---

> ### Author Rebuttal · Authors · 2025-04-01
>
> We thank the reviewer for their comments, which we discuss below.
>
> The reviewer is perfectly right to point out that the choice of edge weights influences the spectral properties of the graph, which impact the robustness criterion. It is thus an interesting research direction to develop a method that maximizes the spectral gap of the *honest subgraph* through a robust protocol, agnostic to the position of the Byzantine nodes.
>
> Meanwhile, it is still possible to build on existing methods, such as the Metropolis choice of weights:
> 1. Given an upper bound on the number of Byzantine nodes $f$, one can have a *local upper bound* on the weight of Byzantine neighbors using $b_i = f \max_{j \in n(i)} w_{ij}$. This can be combined with Metropolis weights to have a robust initialization of the matrix weights. Indeed, as pointed out by [He et al., 2023], defining $w_{ij}$ as $\frac{1}{max(d_i, d_j) + 1}$ is robust: if $i$ is honest and $j$ Byzantine, then $j$ cannot force $i$ to assign a weight greater than $1/(d_i+1)$ to the edge $(i,j)$. In this setting, a local bound of the Byzantine weight can be taken as $b_i=f/(d_i+1)$.
> 2. Having a local upper bound on the number of Byzantine weights $b_i \ge \sum_{j \in n_B(i)} w_{ij}$ is sufficient to implement the robust aggregation rules. For instance, each node $i \in H$ chooses the local clipping threshold based on this local $b_i$. Using such a local upper bound of the weights of Byzantines leads to the same guarantees, with $b = \max_{i \in H} b_i.$
>
> Does our response answer your question?
>
> [He et al., 2023] Byzantine-robust decentralized learning via clippedgossip, He et al. 2023

---

### Official Review · Reviewer_UczM · 2025-03-14

**Overall Recommendation:** 3

**Summary:**

This paper presents a general framework for robust decentralized averaging over sparse communication graphs, providing tight convergence guarantees for various robust summation rules. The authors then investigate the so-called theoretical breakdown: the maximum number of Byzantine nodes an algorithm can tolerate, and demonstrate that some existing approaches like NNA fail earlier. Additionally, they introduce the Spectral Heterogeneity Attack, which leverages graph topology to compromise robustness in sparse networks

**Claims And Evidence:**

A bit lack of experimental evidence.

**Essential References Not Discussed:**

N/A

**Experimental Designs Or Analyses:**

Yes

**Methods And Evaluation Criteria:**

Yes

**Other Comments Or Suggestions:**

N/A

**Other Strengths And Weaknesses:**

Pros:

- Extending robust aggregation from fully connected graphs to sparse graphs is an interesting contribution.

- The unified framework for analyzing theoretical breakdown points is important, especially since it can incorporate some existing well-known approaches as special cases for analysis.

- Rigorous theoretical analysis is provided.

Cons:

- The experiments are relatively simple, focusing only on MNIST and CIFAR-10.

- The paper is somewhat difficult to follow.

**Questions For Authors:**

Would the proposed approach generalize to other decentralized protocols beyond gossip-based ones?

**Relation To Broader Scientific Literature:**

N/A

**Theoretical Claims:**

I do not check all proofs. But the parts I have checked are right.

---

> ### Author Rebuttal · Authors · 2025-04-01
>
> We thank the reviewer for their comments, which we discuss below.
>
> # Experiments
>
> As requested by Reviewers CF8P, UczM, and EynP, we performed additional experiments, which are available here https://anonymous.4open.science/r/rebutal_files-342B/.
>
> We provide 2 additional experiments on Erdös Renyi graphs: one on an averaging task, and one on a CNN trained on MNIST. We additionally compare ourselves with IOS algorithm [3], thus we provide the previous experiments on MNIST as well.
>
> ## Setting
> **Erdős-Rényi** The subgraph of honest nodes is sampled as a random Erdos Renyi graph with 20 honest nodes, each of them being always adjacent to 4 Byzantine nodes. For each seed, 12 different values of $p \in [0.25,1]$ are tested, where p denotes the probability for each edge to exist. We plot the links between the algebraic connectivity of the graph ($\mu_2$, the second smallest eigenvalue of the unitary weighted Laplacian) and the loss. The tasks tested on this graph are:
>    a. **Averaging task**: Nodes' parameters are initialized using a $N(0,I_5)$ distribution. Nodes perform (robust) gossip communication iterations. We conduct experiments on 6 different seeds, and curves are smoothed using a moving average of size 4.
>    b. **MNIST**: The model taken is the same CNN as in the previous experiments on MNIST in the paper. The heterogeneity among nodes has been increased to emphasize the importance of communication, from $\alpha=5$ to $\alpha=1$ (cf. Dirichlet sampling in [3]). Experiments are conducted on one seed.
>
> For clarity, we changed the legend of CS$_{He}$-RG to *ClippedGossip*, even though they both correspond to the same implementable version of ClippedGossip from [1], unsupported by theory.
>
> ## Analysis of the Experiments
>
> ### Averaging Task on Erdős-Rényi
>
> We provide, for each algorithm, the largest value of $\mu_2$ for which it is non-robust and the attack that makes the algorithm break:
> 1. CS$_{ours}$-RG breaks at $\mu_2 \approx 5.5$ with ALIE and FOE attacks.
> 2. ClippedGossip breaks at $\mu_2 \approx 10$ with ALIE attack.
> 3. IOS breaks at $\mu_2 \approx 16$ with FOE attack.
> 4. GTS-RG breaks at $\mu_2 \approx 10$ with SpH attack.
>
> In this setting, $CS_{ours}$-RG is the algorithm with the best breakdown point among the compared methods. Interestingly, $CS_{ours}$-RG is robust for a connectivity $2$ times smaller than GTS-RG, as our theory predict it.
>
> ### MNIST Experiments on Erdős-Rényi
>
> We notice that:
> 1. CS$_{ours}$-RG keeps good performances in this setting while other methods break, showing a better performance than the (worst-case) one predicted in our theory.
> 2. SpH makes IOS break with larger connectivity than other rules. SpH is just as efficient as Dissensus on ClippedGossip, while Dissensus makes GTS-RG break with a larger connectivity.
> ### IOS on MNIST: Previous Experiment
>
> IOS performances are similar to that of GTS-RG: its breakdown point is slightly better than GTS-RG but still worse than CS$_{ours}$-RG. We notice that SpH is very effective against IOS as well in this setting, in the sense that it makes IOS break earlier than other attacks.
>
> # More Involved Datasets
>
> We conducted our experiments on MNIST and CIFAR-10 datasets as those are standard benchmarks in the decentralized robustness literature (see e.g., [He et al., 2023] and [Farhadkani et al., 2022]). Furthermore, the initial experiments on MNIST require approximately 4 days to be performed: indeed, we train simultaneously 26 models for 10 different values of $b$, on 4 different attacks and 3 different algorithms, each experiment being initialized with 5 different seeds. This multiplication of configurations and models to train significantly increases the computational burden of experiments.
>
> As our contributions focus on the averaging sub-problem (the rest is just a direct consequence of the better averaging properties), we provided learning experiments mostly to illustrate our theory, and show how better aggregations lead to better decentralized-SGD algorithms. Our goal was to provide clear theoretical foundations for decentralized optimization, not a large-scale benchmark of current solutions.
>
> # Generalization to Other Decentralized Protocols
>
> Our analysis focuses on the gossip protocol, but it may be of interest to other protocols. Any protocol that, at some point, performs node-level averages of other parameters in the network could apply some $(b,\rho)$-robust summand instead of plain averaging and use our analysis to provide guarantees on the robustness of their algorithm. Note that averaging is particularly relevant, due to the average structure of the ERM problem.
>
>
> [1] Byzantine-robust decentralized learning via clippedgossip, He et al 2023
> [2] Byzantine-resilient decentralized stochastic optimization with robust aggregation
> rules, Wu et al 2023
> [3] Robust collaborative learning with linear gradient overhead, Farhadkhani et al 2023
> [4]  Bridge: Byzantine- resilient decentralized gradient descent. Fang et al 2022

---

### Official Review · Reviewer_CF8P · 2025-03-18

**Overall Recommendation:** 3

**Summary:**

This paper studies Byzantine robust decentralized optimization with a focus on breakdown point. The authors propose a unified method  F-RG, and a new algorithm $CS_{ours}-RG$ adapted for sparse communication networks, both of which have near-optimal breakdown point. Under the proposed $(b, \rho)$-robustness condition, and bounded weight on Byzantine nodes, $r$-robustness and convergence guarantees for decentralized gradient methods are provided for F-RG. Furthermore, the authors propose some new attacks that tested effective in MNIST and CIFAR10 classification tasks.

# post rebuttal update
I thank the authors for the rebuttal. The authors corrected their statement in corollary 4.7, and the updated claim on the spectral limit of $r$-robustness looks great to me. The added discussions on why the proposed sufficient condition is weaker and the final residual errors make the contributions more clear. Thus, I chose to raise my initial score.

**Claims And Evidence:**

The authors provided detailed analysis for the theoretical claims and clear experimental evaluations.

**Essential References Not Discussed:**

N.A.

**Experimental Designs Or Analyses:**

More counterparts besides RG can be added for comparisons, such as the BRIDGE methods in this work: https://arxiv.org/abs/1908.08098

**Methods And Evaluation Criteria:**

The proposed methods are well motivated, and are tested on realistic classification tasks.

**Other Comments Or Suggestions:**

N.A.

**Other Strengths And Weaknesses:**

Strengths:
1. This paper presents a nice characterization of graph(weights) condition, and algorithm condition to ensure byzantine robustness (per my understanding and correct me if I were wrong, it seems that definition 2.1 and 2.2 constitute a sufficient condition for byzantine-robust averaging and gradient methods.  This is nice in the sense that the condition is imposed on the weight instead of the fraction of byzantine agents, which is more intrinsic per my understanding.
2. The paper presents a breakdown analysis for this sufficient condition to hold. The writing is kind of misleading though, since not every byzantine robust decentralized method falls in to the pursuit of the proposed framework, and thus the breakdown point limit is not fundamental in general.
3. Specific algorithms are designed that satisfies the sufficient condition and theoretical guarantees are presented. Attacks are developed, and the proposed algorithms are byzantine robust.

Weakness:
1. The proposed $r$-robustness is very similar to the reduction property used in Farhadkhani et al., 2023. It would be nice if the authors can differentiate from it.
2. The proposed framework does not demonstrate improvement theoretically.

**Questions For Authors:**

N.A.

**Relation To Broader Scientific Literature:**

N.A.

**Theoretical Claims:**

I did not check the proofs of these theorems, but I have some concerns:
1. The theoretical claim in Corollary 4.7 seems to be too good to be true, outperforming the best rate in non-adversarial centralized rate $O(1/\sqrt{T})$.
2. It lacks a detailed comparison with related works, like is this rate better than some previous rates, or this analysis works for a broader class of aggregators or graphs, or the assumptions are relaxed. A detailed comparison would help us to better understand the contributions of these theoretical results.

---

> ### Author Rebuttal · Authors · 2025-04-01
>
> We thank the reviewer for their detailed comments, which we discuss below.
> # Theoretical Improvements
> We respectfully but firmly disagree on the main weakness stated, i.e. our framework does not demonstrate theoretical improvement.
>
> The major theoretical improvements of our framework rely on the tight analysis of the *breakdown point*, which are compared with previous papers in the paragraph *near optimal breakdown point* of Section 3.4.
>
> The table below summarizes the main differences with previous works, with details afterwards.
> |Paper|Algorithm|Heterogeneous losses|Implementable|Generic Analysis|Breakdown assumption|
> |-|-|-|-|-|-|
> |[1]|ClippedGossip|Yes|**No**|No|$b/\mu_{\min}\le \gamma/8$|
> |[2]|NNA|Yes|Only on **fully-connected graphs**|None for $(\alpha,\lambda)$-reduction|$b/H\le 10$|
> |[3]|IOS|Yes|Yes|Yes|$b/\mu_{\min}\le O({1}/{\sqrt{H}})$|
> |[4]|BRIDGE-T|**No**|Yes|No|NA|
> |Ours|CS$_{ours}$-RG (F-RG)|Yes|Yes|Yes|$b/\mu_{\min} \le 1/4$|
>
> We emphasize that:
> - [1] relies on a *non implementable clipping rule* that requires prior knowledge of the neighbors' identity (Byzantine or honest). Yet we have better guarantees: their breakdown point is suboptimal by a significant factor $\gamma$ (the graph's spectral gap) shrinking with the lack of connectivity of the graph.
> - [3]'s breakdown point is highly suboptimal: the proportion of tolerated Byzantine nodes goes to 0 when the number of nodes increases.
> - [2] only considers fully connected networks, still we provide tighter breakdown guarantees: they show NNA tolerates up to 1/11 of Byzantines, while our method tolerates 1/5th. Moreover we show NNA tolerates up to 1/9th. Altough they show multiple steps of NNA satisfy $(\alpha,\lambda)$-reduction with 1/5 of Byzantine nodes, enforcing $\lambda<1$ (required to be r-robust) in their guarantees leads to tolerate only $O(1/\sqrt{H})$.
> # Corollary 4.7 Too Good to Be True
> We clarify here the notation $Var_H(x^t)$, which converges in $O(1/T)$.
> Recall that $Var_{H}(x^t)=1/H\sum_{i\in H}|x_i^t-\bar{x}_H^t|^2$.
>
> It thus quantifies how far honest parameters are from each other, called 'variance among honest nodes'. This may converge to 0 quicker than $O(1/T)$: for an optimization step size $\eta_{op}=0$, the algorithm boils down to (robust) gossip averaging, and the variance converges linearly to 0. Crucially, $Var_{H}(x)$ **does not** correspond to a stochastic variance due to SGD.
>
> PS: Cor. 4.7 statement had a typo: the expectation operator was missing, it might have driven the confusion. Exact result is $E[Var_{H}(x^T)]\in O((1+\zeta^2/\sigma^2)/T)$, where the expectation is taken over SGD randomness.
> # Experiments
> We provide additional experiments (  https://anonymous.4open.science/r/rebutal_files-342B/. ), which are described in Reviewer GrPa's answer.
> # Breakdown Point Limit is Not Fundamental
> We highlight that Thm 3.5 only assumptions are:
> - Algorithm is decentralized, nodes communicate within a graph
> - Algorithm is $r$-robust on this graph
> - Byzantine nodes are unknown.
>
> Thus we make very few assumptions on the nature of the algorithms involved. The r-robustness assumption is only used as a *definition* of a robust algorithm. Crucially, Thm 3.5 does not apply only to F-RG framework.
>
> Hence we do not fully understand the remark on the lack of generality of our breakdown point limit. Would the reviewer know of a method not respecting this limit?
>
> To clarify this section, we will highlight the assumptions of this breakdown point limit more precisely and change the section's title to *Fundamental Limits of r-Robust Algorithms*.
> # Linking $(\alpha,\lambda)$-reduction and $r$-robustness
> We agree $(\alpha,\lambda)$-reduction and $r$-robustness are closely related quantities. While $(\alpha,\lambda)$-reduction measures with two different constants how the *bias* $|\bar{x_i}^t-\bar{x_H}^0|$ and the variance $H^{-1}\sum_{i\in H}\|x_i^t-\bar{x}_H^t\|^2$ evolve during a step of communication, our definition of r-robustness criterion directly quantifies the  of the *mean square error*, i.e. the bias + the variance. Both are linked using:
> 1. $(\alpha,\lambda)$-reduction implies r-robustness with $r=\alpha+\lambda$
> 2. r-robustness implies $(\alpha,\lambda)$-reduction with $\alpha=\lambda=r$
>
> The notion of r-robustness aims at defining what it means to be a "robust decentralized communication algorithm". Indeed, r-robustness with r<1 expresses that "nodes benefit from communicating" more directly than enforcing both $\alpha<1$ and $\lambda<1$. Moreover, the results provided are tighter with this notion of robustness.
> # Conclusion
> We hope that we have adequately addressed all concerns and that these revisions will reflect positively in the reviewer’s final assessment. Otherwise, we would be delighted to engage in further discussion.
> [1] He et al. Arxiv 2023
> [2] Wu et al. IEEE 2023
> [3] Farhadkhani et al. ICML 2023
> [4] Fang et al. IEEE 2022

---

> > ### Comment · Reviewer_CF8P · 2025-04-04
> >
> > I thank the authors for their responses.
> >
> > 1. It would be great if the authors can incorporate the comparisons with other sufficient conditions. That being said, the paper's contribution in proposing a weaker sufficient condition to ensure convergence under Byzantine attack is great, which I also stated in the strengths. In addition, what I meant in my original review is that whether the convergence bound has better dependence on $b$ or $\mu_{\min}$, or on $T$, if you can compare with bounds in other papers. In particular, does the residual error match with existing lower bound in the literature.
> >
> > 2. By saying Corollary 4.7 Too Good to Be True, I mean if you divide both sides by $T$, you can obtain a rate $1/T$, this is even faster than the best rate without Byzantine attack, i.e., $1/\sqrt{T}$ for general non-convex smooth functions. Maybe you missed something somewhere.
> >
> > 3. It would be great if the authors can clarify these presented relations of these two nearly equivalent notions with constant level differences.  It seemed that the authors directly rely on existing bounds built on $(\alpha, \lambda)$ notion to derive Corollary 4.7. Thus, the major contribution of this work would not be in this part, and likely be in deriving a weaker sufficient condition.
> >
> > 4. My comment on 'fundamental limit for decentralized communication schemes' is due to that $r$-robustness or a related $(\alpha, \lambda)$ reduction property, are a good characterization of sufficient conditions for Byzantine robustness. However, seeing a claim such as 'fundamental', I would expect there are some arguments to show that this limit is a necessary condition for objective such as consensus under Byzantine attacks, the current analysis is clearly not along that direction. I appreciated the value of this breakdown point analysis, but stating that it is fundamental may be misleading.

---

> > > ### Author Response · Authors · 2025-04-07
> > >
> > > # 1) Corollary 4.7 too good to be true
> > >
> > > We would like to apologize sincerely to the reviewer: we indeed made a typo in the statement of Corollary 4.7, which should have been stated as follows.
> > >
> > > The case of one step of F-RG step between each optimization step writes, for $i$ honest:
> > >
> > > \begin{equation}
> > > \frac{1}{T}\sum_{t=1}^T E\|\nabla f_{H}(x_i^t)\|^2 = O\left( \frac{L\sigma}{\gamma(1-\delta)\sqrt{T}} + \frac{\zeta^2}{\gamma^2(1-\delta)^2}\right).
> > > \end{equation}
> > >
> > > While the multi-communication steps derivation yields:
> > > $$
> > > \frac{1}{T}\sum_{t=1}^T E\|\nabla f_{H}(x_i^t)\|^2 = O\left( L\sigma\sqrt{\frac{\delta}{\gamma(1-\delta)^2T}} + \frac{\delta}{\gamma(1-\delta)^2}\zeta^2\right).
> > > $$
> > >
> > > NB: We assumed $\delta \ge 1/H$ in both results and $\gamma \ll 1$ in the single communication step result.
> > >
> > > # 2) Convergence bounds w.r.t. existing ones
> > >
> > > ## Residual error in D-SGD
> > >
> > > Up to our knowledge, existing lower bounds on the error due to Byzantine corruption were designed in the non-decentralized case, and thus do not involve graph-related quantities such as the spectral gap.
> > >
> > > The lower bound proven in [5] shows that there exist configurations that satisfy the heterogeneity part of Assumption 4.6, and on which no algorithm can have a better error than $\Omega(\frac{b}{H-b}\zeta^2)$, where (here) $b$ is the number of Byzantines among the $H+b$ workers.
> > >
> > > **Fully connected case.**
> > > 1. Our bound for the multiple communication case matches this lower bound on fully-connected graphs (i.e. when $\gamma =1$) up to a factor $\frac{1}{1-\delta}$, since here $\delta= \Theta(b/H)$.
> > > 2. The result we provided for the single communication step case was simplified by assuming that $\gamma \ll 1$ (cf. Appendix F, Equation 13) for clarity and space constraints. If, instead, we plug $\gamma =1$ in Equation 13, the asymptotic error achieved is the same as previously, i.e $\frac{\delta}{(1-\delta)^2}\zeta^2$.
> > >
> > > **Sparse graph case.** Our asymptotic error (in the multiple communication step derivation) is sub-optimal with respect to the lower bound in the non-decentralized case by a factor $(\gamma(1-\delta))^{-1}$. It is an open question whether this is optimal in the sparse graph regime or not.
> > > 1. The convergence results of [1] show the same asymptotic error as ours (if we ignore the $1-\delta$ terms, which are non-significant away from the breakdown point).
> > > 2. On the opposite, [2] have an asymptotic error of
> > > $$
> > > O\left((b/\mu_{min})^2 \sigma^2 \gamma^2 H \left(\frac{1 - \gamma(1- b/\mu_{min} \sqrt{H})}{\gamma^3(1- b/\mu_{min} \sqrt{H})^3} + 1 + \zeta^2/\sigma^2\right)\right).
> > > $$
> > > which is worse than ours: for instance, in fully connected graphs and under $\sigma^2=0$, this error boils down to $\frac{b^2}{H} \zeta^2$, which is suboptimal by a factor $b$, here the number of Byzantines workers.
> > >
> > > ## Convergence rate in D-SGD
> > > Comparison of the multiple communication steps convergence rate to existing cones:
> > >
> > > 1. We rely on the proof of [3] for our bounds, as such we both have the same convergence rate on fully-connected graphs.
> > > 2.  We slightly improve over the convergence rate of [1] by a factor $\sqrt{L}$.
> > > NB: We compare ourselves to their detailed convergence result in their appendix, since the simplified one of the main part of the article has a typo (an additional $1/H$ factor in front of their convergence rate that does not appear in the appendix: neither in the theorem III statement nor its proof).
> > >
> > > 3. Even though [2] presents significantly worse asymptotic error and breakdown point, their convergence rate is $O(\sqrt{\sigma^2/(HT)})$, which is better than ours.
> > >
> > > We will add this discussion to our work.
> > >
> > > # 3 Clarification
> > >
> > > We fully agree with the reviewer that Corollary 4.7 is not a contribution of our work. We stated this corollary
> > > 1. to show that our results can be conveniently combined with existing work to derive efficient D-SGD algorithms;
> > > 2. to compare ourselves theoretically and experimentally with existing papers, which generally focus on Byzantine's robust D-SGD framework.
> > >
> > > For these reasons, we have not called it a “theorem”, but a “corollary” to emphasize that it is a mere consequence of our results.
> > >
> > >
> > >
> > > # 4 Fundamental limits
> > >
> > > We agree that this title is indeed misleading if "fundamental limits" suggest that we prove a necessary condition. We thank the reviewer for raising this point, and we will change the title to *Spectral limit of r-robust algorithms*.
> > >
> > > ## Additional references
> > >
> > > [5] Robust Distributed Learning: Tight Error Bounds and Breakdown Point under Data Heterogeneity, Allouha et al. 2023

---

### Decision · Program_Chairs · 2025-05-01

**Decision:**

Accept (poster)

**Comment:**

Reviewers agree that the work makes important theoretical contribution to decentralized robust optimization. Reviewers also find the unified framework useful, and strongly suggest authors to incorporate key points in rebuttal into the revision.